# Basin-wide sea level coherency in the tropical Indian Ocean driven by Madden–Julian Oscillation

B. Rohith[1,2], Arya Paul [1], Fabien Durand[3], Laurent Testut [3,4], S. Prerna[1], M. Afroosa[1,5], S.S.V.S. Ramakrishna[2] & S.S.C. Shenoi[1]

Changes in sea level may be attributed either to barotropic (involving the entire water column) or baroclinic processes (governed by stratification). It has been widely accepted that barotropic sea level changes in the tropics are insignificant at intraseasonal time scales (periods of 30–80 days). Based on bottom pressure records, we present evidence for significant basin-wide barotropic sea level variability in the tropical Indian Ocean during December–April with standard deviations amounting to ~30–60% of the standard deviation in total intraseasonal sea level variability. The origin of this variability is linked to a small patch of wind over the Eastern Indian Ocean, associated with boreal winter Madden–Julian Oscillations (MJO). These large fluctuations are likely to play a prominent role in the intraseasonal sea level and mass budgets. Because of their much faster propagation than baroclinic processes, they allow the basin to adjust to climatic perturbations much more rapidly than was previously thought.

---

[1] Indian National Centre for Ocean Information Services (INCOIS), Pragathi Nagar, Hyderabad 500090, India. [2] Department of Meteorology and Oceanography, Andhra University, Visakhapatnam 530003, India. [3] LEGOS, Université de Toulouse, IRD, CNRS, CNES, UPS, Toulouse 31400, France. [4] LIENSs, Université de La Rochelle, CNRS, La Rochelle 17000, France. [5] School of Ocean Sciences & Technology, Kerala University of Fisheries and Ocean Studies, Panangad, Cochin 682506, India. Correspondence and requests for materials should be addressed to A.P. (email: aryapaul@incois.gov.in) or to S.S.C.S. (email: shenoi@incois.gov.in)

At intraseasonal time scales, the response of the ocean to wind stress, whose spatial extent is limited to 100 km, is mostly trapped near the ocean surface[1] and the response is primarily baroclinic. However, if the extent of forcing exceeds a few 100 km and the forcing is strong enough, the response can reach the ocean bottom, triggering a barotropic response[1]. Until now, the intraseasonal barotropic response of the ocean to external fluxes has been assumed to be negligible in tropical regions, based on altimetry studies[2] and analytical exercises[1] that relied on approximations of linearity, quasi-geostrophy and negligible bottom slope. This assumption, if not valid, has important consequences for the interpretation of altimeter-derived sea level anomalies (SLA). Considering that the barotropic variability is unimportant in the tropics, the intraseasonal SLA in the tropical oceans is interpreted as a manifestation of baroclinic processes[3,4] alone. However, it is necessary to examine the barotropic components because the processes attributed to baroclinic modes take place against the background of barotropic adjustments. In particular, the existence of Madden–Julian Oscillations (MJO)[5,6], involving large-scale (≫1000 km) coherent atmospheric modulations at intraseasonal time scales in the Indo-Pacific region, presents a theoretical possibility of triggering barotropic motions in tropical oceans. Here, we investigate whether there are significant large scale barotropic SLA variabilities in the tropical ocean and, if so, where they originate and what are their spatiotemporal scales.

Signatures of significant barotropic SLA are primarily reflected in the bottom pressure variability. Indeed, even though the bottom pressure anomalies are susceptible to baroclinic processes, this effect is negligible at large depths (typically outside the continental shelves and slope areas) in a flat ocean bottom[7–10]. Hence, it is appropriate to look for barotropic signatures in bottom pressure recorders (BPR) anchored at large depths. In this work, we use the measurements from three BPRs in the tropical Indian Ocean (TIO) to identify the barotropic SLA, and an ocean general circulation model (OGCM) to identify their origin.

Here, we show that large wind anomalies over the north-west Australian basin (NWAB), associated with boreal winter MJOs, cause a significant basin-wide barotropic sea level response in the TIO at intraseasonal time scales. This novel feature is deciphered using BPRs, Gravity Recovery and Climate Experiment[11] (GRACE) and an OGCM. The dynamics of this large scale coherent sea level response has been discussed using a set of sensitivity experiments. Interestingly, stratification plays a prominent role in explaining the basin-wide variability—a characteristic counter-intuitive to barotropic dynamics. This new finding challenges the existing notion that barotropic dynamics is of little significance in the tropical oceans and hence demands greater caution in the interpretation of intraseasonal SLA.

## Results

**Observation of barotropic sea level variability**. The BPRs record ocean bottom pressure, which is expressed as an equivalent water depth (EWD) once scaled by the mean ocean density and the acceleration due to gravity. We have processed (see Methods, Supplementary Figure 1) and analysed 6-year long de-tided EWD anomaly time series from three BPRs (Fig. 1) located in Bay of Bengal, Arabian Sea and NWAB. The BPRs are hereafter referred to as BP-BoB (88.80° E, 6.25° N, 3793 m), BP-AS (65.33° E, 20.80° N, 2612 m) and BP-NWAB (117.94° E, 15.02° S, 5664 m) (Fig. 2b).

The magnitude of EWD variability in the TIO amounts to ~4–6 cm (Fig. 1a) and is most pronounced at BP-NWAB, which lies on the path of boreal winter MJO. In addition, all three BPRs appear to fluctuate coherently during December–April, even though they are separated by thousands of kilometres. Wavelet analyses of the time series from the three BPRs suggest that most of the energy is in the 30–100 days band at all the three locations (Fig. 1b–d) during December–April. The cross-wavelet transform, a method that measures the degree of coherency as a function of frequency, indicates that the coherency between BP-BoB and BP-AS is most pronounced in the 30–80 days band (Fig. 1e) and during the boreal winter months of 2011–2012, 2012–2013 and 2014–2015 (Fig. 1e). A similar coherency is observed between BP-BoB and BP-NWAB (Fig. 1f) during the boreal winter months. During 2013–2014, the coherency between BP-BoB and BP-AS, as well as BP-BoB and BP-NWAB, is insignificant. This is also evident in Fig. 1a where the EWD anomaly fluctuations during December–April of 2013–2014 appear rather incoherent and their magnitudes are considerably reduced, leading to a comparatively smaller signal-to-noise ratio. Overall, the results suggest a large-scale coherent behaviour during December–April of each year, except during 2013–2014.

Hence, EWD anomaly time series were band-pass filtered to extract the signals between 30 and 80 days (hereafter referred to as the intraseasonal band) using a Lanczos filter (Fig. 2a). The increase in standard deviation in intraseasonal EWD anomaly during December–April with respect to the rest of the months is largest at BP-NWAB (57%) and smallest at BP-AS (15%). Even though the three BPRs coherently fluctuate at intraseasonal time scale, there are instances when BP-AS looks to respond to other frequencies as well—may be because it is located close to the coast and its EWD is prone to respond to the baroclinic processes. The variability in the intraseasonal EWD anomaly reached up to ~8 cm peak to peak. We henceforth restrict our attention to the intraseasonal band and December–April months unless otherwise mentioned. The absence of significant phase lag in EWD anomaly at three widely separated locations (Fig. 2a) rules out the dominant involvement of slow baroclinic processes. Indeed, baroclinic processes might have been relevant if there would have existed a large-scale intraseasonal forcing mechanism in the TIO during December–April encompassing all the three BPR locations. However, no such large-scale forcing has ever been reported. To further explore the possibility of basin-wide EWD variability, the EWD anomaly at BP-BoB was correlated with the EWD anomaly derived from GRACE (Fig. 2c). The broad, positive correlation pattern confirmed that the EWD oscillations are basin-wide. The relative contribution to EWD variability through baroclinic processes[10] derived from an OGCM (MOM4p1) in the TIO is negligible at intraseasonal time scales (see Methods, Supplementary Figure 2). These results point towards a prominent barotropic dynamics driving the observed EWD variability. Hence, we identify EWD variability with barotropic SLA in the following. This is the first time such a basin-wide variability in the barotropic SLA is being reported in the TIO. The standard deviation of barotropic SLA at intraseasonal time scale amounts to about 62%, 36% and 28% of the standard deviation seen in the intraseasonal SLA from AVISO altimetry during December–April at BP-AS, BP-NWAB and BP-BoB, respectively (see Supplementary Figure 3). This is very significant and demands attention.

**The source of coherent barotropic SLA**. The observed basin-wide synchronous oscillations in barotropic SLA are unlikely to be produced by atmospheric pressure because (i) the effect of atmospheric pressure on open ocean bottom pressure is negligible at time scales longer than ~3 days[12] and (ii) the dynamic response of barotropic SLA to atmospheric pressure is an order smaller than that induced by the wind[12,13]. Basin-wide oscillations in barotropic SLA are also unlikely to be produced by coherent local winds because the spatial scales of these winds are

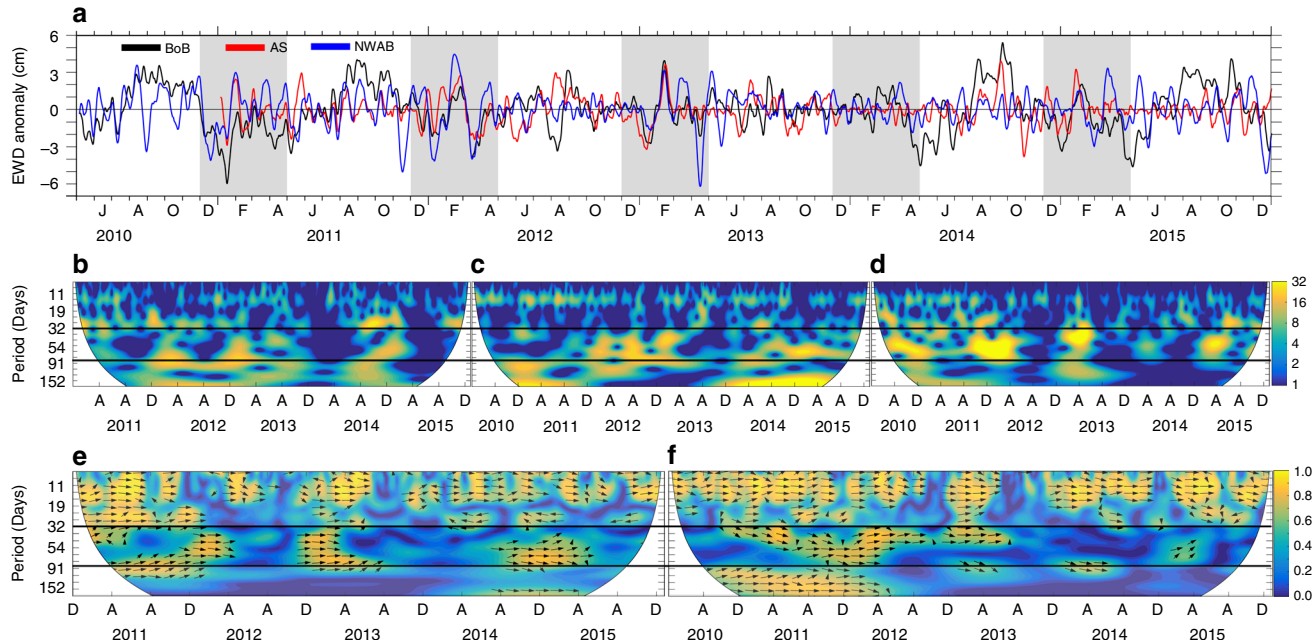

**Fig. 1** Oscillations of EWD anomaly observed by the BPRs and their energy coherency characteristics. **a** Time series of EWD anomaly (10-day smoothed for clarity) at BP-BoB (black), BP-AS (red) and BP-NWAB (blue). Light grey sections underline the seasonal period of interest, from December through April. $y = 0$ line is plotted for clarity. J, A, O, D, F, A in the x-axis labels denote June, August, October, December, February and April, respectively. The continuous wavelet power spectrum at **b** BP-AS, **c** BP-BoB and **d** BP-NWAB. Cross-wavelet transform of the standardised **e** BP-BoB and BP-AS and **f** BP-BoB and BP-NWAB time series of EWD anomaly. The relative phase coherency is shown as arrows (in-phase pointing right, out of phase pointing left). The two-solid horizontal black lines in (**b**)–(**f**) enclose the 30–80 days band whereas A, A, D in the x-axis labels denote April, August and December, respectively. The cone of influence where edge effects might distort the picture is blanked out. BPR locations are shown in Fig. 2b

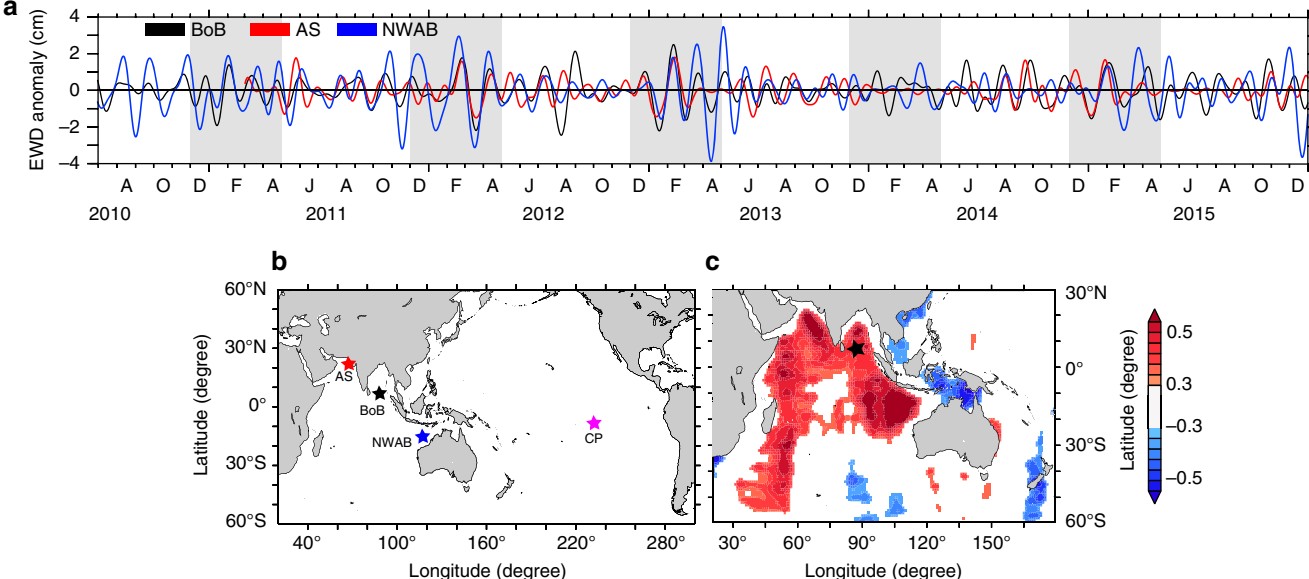

**Fig. 2** EWD anomaly time series and its correlation with space-borne gravimetry measurements. **a** Time series of the intraseasonal band (30–80 days) EWD anomaly at BP-AS (red), BP-BoB (black) and BP-NWAB (blue). Light grey sections underline the seasonal period of interest, from December through April. $y = 0$ line is plotted for clarity. J, A, O, D, F, A in the x-axis labels denote June, August, October, December, February and April, respectively. **b** Location of the BPRs used in this study: Arabian Sea (AS; red), Bay of Bengal (BoB; black), North-West Australian Basin (NWAB; blue) and Central Pacific (CP; pink). **c** Correlation during December–April between observed intraseasonal EWD anomaly at BP-BoB and 10° × 10° spatially smoothed intraseasonal EWD anomaly from GRACE gravimetry (95% significance)

much smaller than the basin. The most likely sources are therefore remotely-forced fast barotropic waves generated by large-scale winds and/or baroclinic–barotropic conversions[14].

Sea level changes due to winds through barotropic processes under linear quasi-geostrophic approximations are driven by $f(\nabla \times \tau/H)$ where $f$ is the Coriolis parameter, $\tau$ is the wind stress and $H$ is the ocean depth[2]. We have computed the spatial correlation between barotropic SLA at BP-BoB and the vertical component of intraseasonal wind forcing $f(\nabla \times \tau/H)$ during December–April (Fig. 3). The correlation is insignificant at the BP-BoB location, ruling out the possibility of local winds as the source of coherent basin-wide barotropic SLA. However, the correlation is significant over a patch of the eastern TIO between 33° S and 2° N (boxed region, Fig. 3; negative as $f$ is negative there), suggesting that the wind stress over NWAB could play a significant role in generating basin-wide oscillations in TIO. This could also be the reason why the barotropic SLA fluctuations are significantly larger at BP-NWAB than at the other two BPR locations (see Fig. 2a). This region also coincides with the region where MJO wind stress anomalies are large in November–April[15,16]. The gradients in underlying topography of the NWAB region transforms the dipolar structure in the correlation map, typically associated with canonical MJOs, into a monopole: the dipole in wind stress curl exist when the underlying topography is flat (see Supplementary Movie 3), and the northern pole collapses once the gradients in topography are considered (see Supplementary Movie 1). The intensity of winter MJO at its various phases[17] of eastward propagation is shown in Fig. 4a. An intense MJO in phases 3–6 in 2011–2012, 2012–2013 and 2014–2015 indicated strong winds over the boxed region[15,16] and corresponds to a strong barotropic SLA in the TIO. On the contrary, the weak MJO in 2013–2014 indicated weak winds and consequently weak barotropic SLA variability (Fig. 2a). The correlation between the daily Real-time Multivariate MJO series 2 (RMM2)[17] and barotropic SLA at BP-BoB/BP-NWAB/BP-AS is 0.63/0.44/0.40 (significant at >97%). The results are qualitatively similar at all the BPR locations and we show the results pertaining to BP-BoB in the rest of the paper.

We have used a three-dimensional OGCM (the MOM4p1 model[18]) to evaluate the capability of the model in reproducing the observed barotropic SLA at BPR locations. The model was forced with wind stress, heat and freshwater fluxes (the UV experiment, Methods). The impact of atmospheric pressure on model EWD anomaly (equivalently termed as barotropic SLA) did not exceed 3 mm of variability in the TIO; hence, neglected in subsequent discussions (Methods, Supplementary Figure 4). The model (the UV experiment) generally reproduced the observed amplitude and phase at BP-BoB (Fig. 4b; black dotted line). The spatial correlation between the observed barotropic SLA and the model barotropic SLA from UV experiment (Fig. 4c) reproduced the pattern observed using gravimetry (Fig. 2c), confirming that the whole TIO fluctuates coherently at intraseasonal time scales during December–April.

Inspired by the correlation with winds (Fig. 3), we ran an experiment confining the wind stress only over the NWAB region (the boxed region in Fig. 3; the NWAB experiment, Methods, Supplementary Table 1). The strong similarity between the observed barotropic SLA and the model barotropic SLA from NWAB experiment, particularly during the strong MJO years of 2011–2012, and 2012–2013 and 2014–2015, indicates that the winds over NWAB play a prominent role in setting up the coherent barotropic SLA fluctuations (Fig. 4b; green curve). The NWAB experiment however did not reproduce the barotropic SLA evolution in December 2010–January 2011, December 2013–January 2014 and December 2014–January 2015 when the MJO was weak in phases 3–6 (Fig. 4a). This suggests that

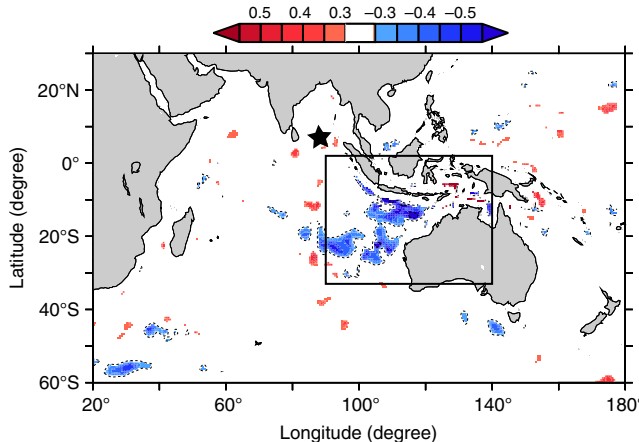

**Fig. 3** The source region of barotropic SLA variability. Correlation between observed intraseasonal barotropic SLA at BP-BoB and the vertical component of intraseasonal $f(\nabla \times \tau/H)$. Wind stress, $\tau$, was taken from NCMRWF for the months of December–April during 2010–2015. $H$ was taken from ref. [33]. Only correlations significant at the 90% level are plotted. The black star indicates the location of BP-BoB

whenever a strong MJO traverses through the NWAB basin, it produces a significant sea level variability in the TIO basin. The variance in the barotropic SLA from UV experiment explained by the NWAB experiment during all the December–April seasons of 2010–2015 at BP-BoB amounts to ~60%. It is striking that a relatively small area (NWAB region) lying along the path of MJO manages to significantly drive sea level over the entire TIO basin at intraseasonal time scales.

Since the observed variability results from horizontal mass fluxes driven by barotropic motions, we have further investigated whether a barotropic model could reproduce the observations. We have repeated the NWAB experiment, this time removing the effects of vertical stratification by imposing uniform temperature and salinity throughout the domain (NWAB-NS experiment, see Supplementary Table 1). This experiment reproduced the phase of the observed signal but the amplitude was reduced by a factor of ~1.5–1.7 (Fig. 4b; cyan curve). To assess the influence of topography, we removed the effects of topographic gradients by imposing a flat ocean bottom everywhere but retained the vertical stratification (NWAB-FB experiment, see Supplementary Table 1). The average depth of the NWAB basin, $H = 3000$ m, was chosen as the depth of flat bottom. Interestingly, the barotropic SLA produced in the NWAB-FB experiment (Fig. 4b; purple curve) is similar in magnitude to those generated in the NWAB-NS experiment, (Fig. 4b; cyan curve), but lower than NWAB experiment by a factor of 1.5–1.7. Expectedly, this scaling factor in the NWAB-FB experiment is strongly sensitive to $H$ (Supplementary Figure 5) because it plays a prominent role in determining the rate of vorticity injection by the wind stress into the ocean. However, this strong sensitivity to $H$ is significantly mitigated if the wind stress is scaled such that the wind forcing term $f(\nabla \times \tau/H)$ remains unchanged. The barotropic response at BP-BoB however remains essentially unaltered (see Supplementary Figure 6). The NWAB, NWAB-NS and NWAB-FB experiments highlight the importance of wind stress at the NWAB, along with the stratification and the topography over the region in controlling the barotropic SLA amplitude. However, since the NWAB experiment could reproduce most of the variability observed in the barotropic SLA over the TIO, we conclude that the NWAB region is instrumental in producing the intraseasonal barotropic SLA over the TIO. In other words,

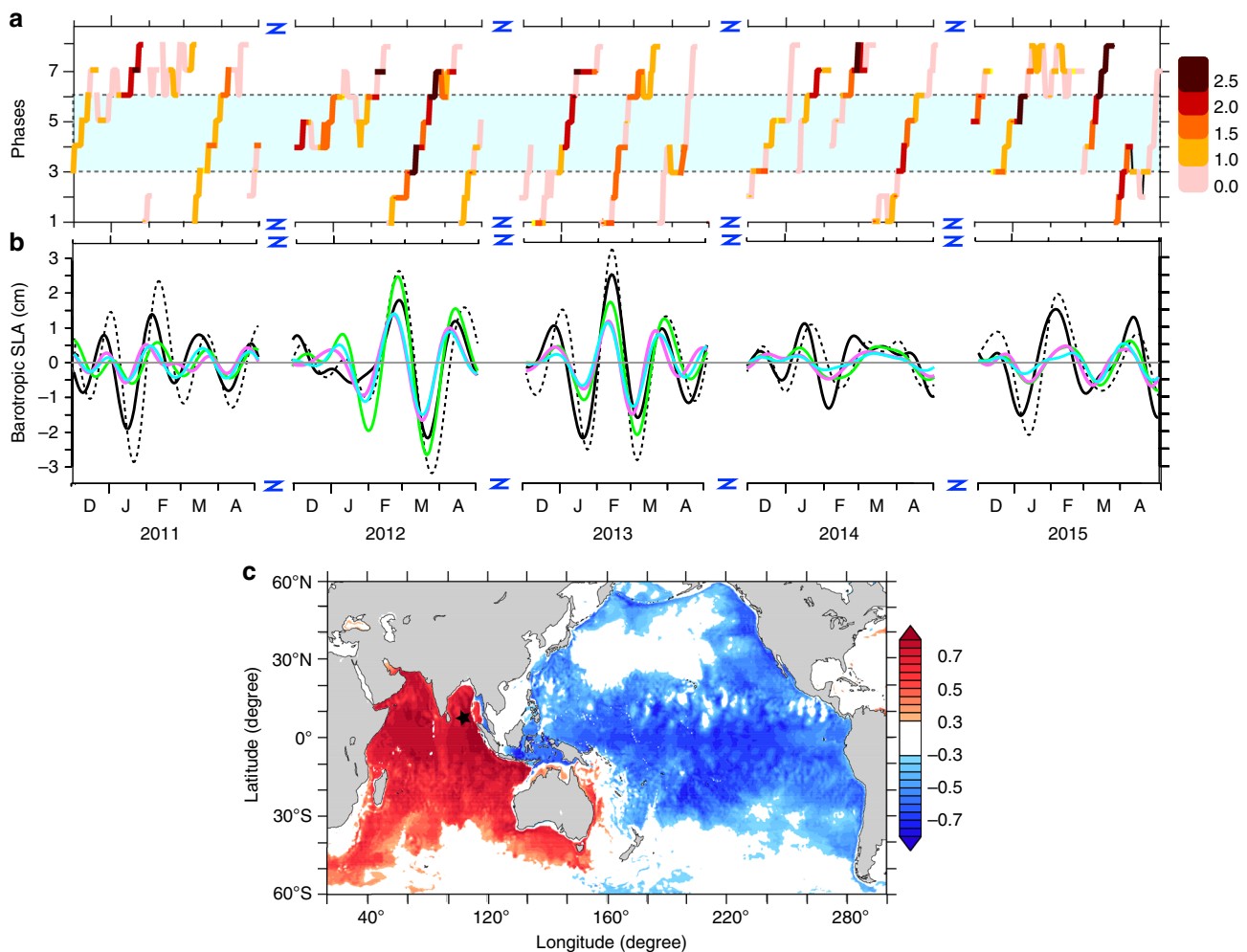

**Fig. 4** MJO phase diagram and comparisons between models and observations. **a** Diagram of amplitude and phase of MJO index[17] (see Methods for details). The boxed region in Fig. 3 witnesses winds due to MJO in its phases 3–6 (i.e. inside dashed lines marked by the cyan patch) (Methods, Supplementary Figure 7). D, J, F, M, A represent December, January, February, March and April, respectively. **b** Intraseasonal barotropic SLA time series at BP-BoB during December–April from observations (black) and the experiments, UV (black dotted line), NWAB (green), NWAB-FB (purple) and NWAB-NS (cyan). $y = 0$ line is plotted for clarity. D, J, F, M, A represent December, January, February, March and April, respectively. **c** Correlation between the intraseasonal barotropic SLA at BP-BoB (black star) and intraseasonal barotropic SLA from the UV experiment during December–April for 2010–2015 (95% significance)

the wind stress in the NWAB region generates the intraseasonal barotropic SLA that gets amplified by a factor of ~1.5–1.7 in a baroclinic ocean with undulating bottom topography. Though the dynamics of this interplay is not yet clear, the joint effect of baroclinicity and relief[19,20] (JEBAR), also known as pycnobathic forcing[21], probably causes the amplitude enhancement.

**Dynamics of barotropic SLA.** To gain more insights into the dynamics driving the basin-wide coherent oscillations, we have animated the hourly barotropic SLA from the NWAB experiment (Supplementary Movie 1) during December 2011–April 2012 which is ostensibly a period of strong MJO. The wind stress over the boxed region generates three kinds of waves. The propagations of these waves are illustrated schematically in Fig. 5.

During each cycle, the wind stress drives out three distinct waves from the box. A positive (negative) wind stress curl drives (i) a fast north-westward propagating wave (red arrow, Fig. 5) with a negative (positive) SLA that spreads across the northern TIO, bumps into the coast of Asia and Africa and reflects back to fill the central Indian Ocean; (ii) a positive (negative) wave (pink

arrow, Fig. 5) propagating southward parallel to the Australian shelf and ultimately turning westward to fill the southern part of the TIO; and (iii) a positive (negative) wave (yellow arrow, Fig. 5), originating at the north-west coast of Australia and propagating southward along the Australian shelf, hugging the west coast. The second and third waves are demarcated by a $\frac{f}{H}$ contour that runs parallel to the west and south coast of Australia. Another control experiment (NWAB-NR experiment; Supplementary Table 1) is performed where the topography outside of NWAB region is gradually flattened over a length scale of 500 km thereby leading to weak bottom slopes and hence mitigates possibilities of baroclinic-to-barotropic conversions in the perimeter zone of the forcing region. It shows that the topography (outside NWAB) plays a prominent role in relaying the barotropic information to the BPR sites. The absence of ridges outside the forcing region in the NWAB-NR experiment allows the planetary wave to travel westward instead of north-westward and the topographic waves to leak into the Southern Indian Ocean thereby decreasing the amplitude of model barotropic SLA by ~23% (see Supplementary Movie 4) compared to the NWAB experiment. In contrast, the

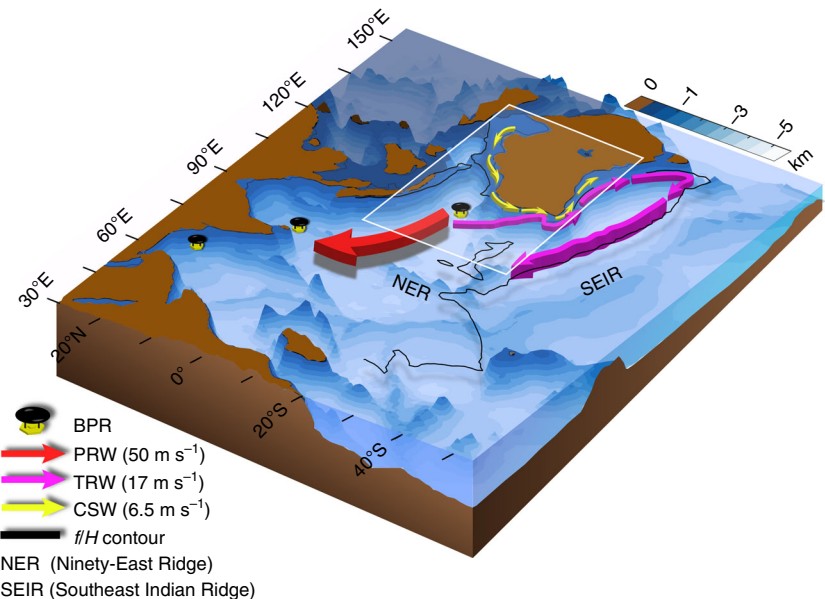

**Fig. 5** Pathways of barotropic waves in the Indian Ocean. Schematic diagram of the propagation of barotropic waves forced by winds over the NWAB (white box). The arrows represent the pathways of barotropic waves. Propagation speeds are inferred from Supplementary Figure 9. The red arrow pointing north-westward originating from NWAB is a planetary Rossby wave (PRW) propagating with a phase speed of 50 m s$^{-1}$. The pink arrow, parallel to the Australian shelf, and exiting the box to the southeast is a topographic Rossby wave (TRW) that follows a $\frac{f}{H}$ contour (black line) propagating with a phase speed of 15 m s$^{-1}$. This wave then turns westward, along the $\frac{f}{H}$ contour. The thin yellow arrow hugging the west coast of Australia is a continental shelf wave (CSW) propagating southward with a speed of 6.5 m s$^{-1}$. Southeast Indian ridge appears to constrain the propagation of barotropic waves to the Southern Indian Ocean. Blue shading represents the bathymetry. The location of the three BPRs is marked using a black-yellow symbol. The Ninety East Ridge (NER) and the Southeast Indian Ridge (SEIR) are also shown

Southeast Indian Ridge in the NWAB experiment acts as a wall and shields the Southern Indian Ocean from the effects of barotropic dynamics originating over NWAB basin thereby facilitating enhanced variability in the barotropic SLA (see Supplementary Movie 1). The absence of Southeast Indian Ridge solely induces a reduction of ~12–16% in the intraseasonal barotropic SLA fluctuation in the BP-BoB location (Supplementary Figure 8a). The Ninety East Ridge running parallel to the western edge of the box, acts as an inhibitor and encourages the wave to pass through the open passage at its northern edge and further propagate westward and north-westward. The Ninety East Ridge however negligibly influence the amplitude of the barotropic SLA in the north Indian Ocean (Supplementary Figure 8b).

We have performed another set of control experiments to understand the nature of these waves (Supplementary Table 1). The first wave (red arrow, Fig. 5) is absent if $f$ is held constant (NWAB-CF experiment; Supplementary Movie 2) suggesting that it is governed by planetary vorticity gradients and is therefore a planetary Rossby wave. The amplitude of intraseasonal barotropic SLA at BP-BoB in NWAB-CF experiment is reduced by ~95% (compared to NWAB experiment) thereby suggesting that these planetary waves are instrumental in facilitating the observed barotropic sea level fluctuations particularly in the north and central Indian Ocean. In the absence of damping, under quasi-geostrophic dynamics, the phase speed of a barotropic planetary Rossby wave forced by a wind patch with a north–south length-scale ($l$) of ~1000 km is $\beta l^2$ [7,22]. Assuming a value of $\beta = 2 \times 10^{-11}$ m$^{-1}$ s$^{-1}$ (corresponding to 15° S), the theoretical phase speed of this planetary Rossby wave forced by the wind patch (~1500 km) in the western side of the box is 45 m s$^{-1}$. The calculated average phase speed from the model is 50 m s$^{-1}$ (Supplementary Figure 9b). The second (pink arrow, Fig. 5)

and third (yellow arrow, Fig. 5) waves do not exist if the ocean bottom is flat (NWAB-FB experiment; Supplementary Movie 3) thereby ruling out the possibility that either wave is a coastal Kelvin wave. The third wave (yellow arrow, Fig. 5), which propagates with an average phase speed of 6.5 m s$^{-1}$ (Supplementary Figure 9d) in the NWAB experiment, is consistent with a continental shelf wave. The second wave travels westward from the southern tip of Australia with a model-derived average phase speed of 17 m s$^{-1}$ (Supplementary Figure 9c). The path followed by this wave is along a constant $\frac{f}{H}$, suggesting that it is a topographic Rossby wave. For inviscid fluids under quasi-geostrophic dynamics, the theoretical speed of a short barotropic topographic Rossby wave travelling westward through a channel of width $L$ and of characteristic depth $D$ is approximately $sgD/fL$, where $s$ is the bottom slope of the channel[23]. The average depth of the ocean at 40° S, where the model speed is evaluated, is ~3000 m, the characteristic width of the channel is ~500 km and the average slope is $s = 2 \times 10^{-2}$, yielding a theoretical speed of ~12.5 m s$^{-1}$, close to the model estimate. The combined effect of the planetary Rossby wave and the topographic Rossby wave results in a basin-wide barotropic sea level fluctuation that ranges from the northern boundary of the Indian Ocean to the Southeast Indian ridge. The continental shelf wave, in contrast, appears to play no role in setting up this basin-wide response.

## Discussion

The intraseasonal variability in the sea level observed at the BPRs appears to have been largely caused by barotropic waves generated by the wind stress associated with the MJO in the NWAB. The findings reported here, in particular the prevalence of large barotropic variability over vast parts of the tropical oceans at intraseasonal time scales and the importance of baroclinic physics

in their generation, should be accounted for while interpreting the signals obtained by space-borne altimetry and gravimetry. The widespread belief that, in the tropics, the intraseasonal SLA derived from altimetry is primarily associated with baroclinic transients therefore requires re-evaluation. Indeed, observed SLA contains a significant fraction of barotropic variability in the TIO in the intraseasonal frequencies during December–April; particularly when the MJO is strong during phases 3–6. These intraseasonal barotropic SLA variabilities, with an amplitude of ~6–8 cm, amount to ~30–60% of the standard deviation of total observed intraseasonal SLA in the TIO.

The draining in (out) of water mass observed in the TIO requires an external source (sink). The boreal winter EWD anomalies observed at a BPR located in the central Pacific Ocean (BP-CP; 125° W, 8.5° S, 4449 m) often appear out-of-phase with the observed barotropic SLA at BP-BoB (Fig. 6), in particular during late 2010, early 2012 and during 2013. This suggests that the Pacific Ocean may be a significant reservoir that supplies (receives) the water mass drained in to (out of) the TIO. Given that, beyond the TIO, the region of influence of the MJO also comprises the Pacific basin, this opens the question of the existence of a similar dynamics as the one described in the present study, acting over the tropical Pacific basin.

The adjustment time scale of sea level to MJO along the rim of the Indian Ocean was estimated to a few weeks to months in previous works[24,25], resulting primarily from long baroclinic Rossby and Kelvin waves propagating at low speeds. Our results, in contrast, show that part of this adjustment takes place over a few tens of hours through these fast barotropic waves.

Any regional model aiming to realistically simulate the full range of Indian Ocean sea level dynamics may account for the dynamics identified in the present study, both barotropic and baroclinic, and may encompass the area of atmospheric forcing identified here. Future studies devoted to sea level variability in the TIO (whether regionally or at the basin scale) or attempting to close the sea level budget should consider the effects reported here while interpreting the intraseasonal signals.

Our model experiments showed that the observed barotropic SLA in TIO is a manifestation of barotropic transients caused by MJO winds and accentuated by the interplay of stratification and topography in the NWAB. This interplay induces a conversion from baroclinic to barotropic transport, i.e. small-scale processes merge onto large-scale processes and hence may explain why linear barotropic vorticity equations, that neglect stratification, cannot explain the vorticity budget. The bathymetry in the NWAB region influences a significant fraction of the observed barotropic SLA in the TIO. The bathymetry outside the NWAB region, however, plays a different role. The Ninety East Ridge steers the planetary wave north-westward whereas the Southeast Indian Ridge acts as a natural boundary and contains the topographic wave from spilling over to the Australian–Antarctic Basin thereby limiting its interference with the large intraseasonal barotropic fluctuations prevalent over there[26,27].

We can expect that these barotropic waves carry significant energy and are likely to play a prominent role in the ocean energy and momentum budget. The velocity of the barotropic flow associated with these waves, estimated from the UV experiment, typically amounts to 2 cm s⁻¹ in the deep ocean regions. As such, these waves may also act as a reservoir of energy for deep ocean intraseasonal variability. The dynamics revealed here raise the possibility of energy exchange between barotropic and baroclinic transients, particularly in regions of large topographic gradients where these vertical modes are prone to interact[21,28].

As the intensity of MJOs is expected to strengthen in a changing climate[29], the future evolution of the dynamics reported here requires to be investigated.

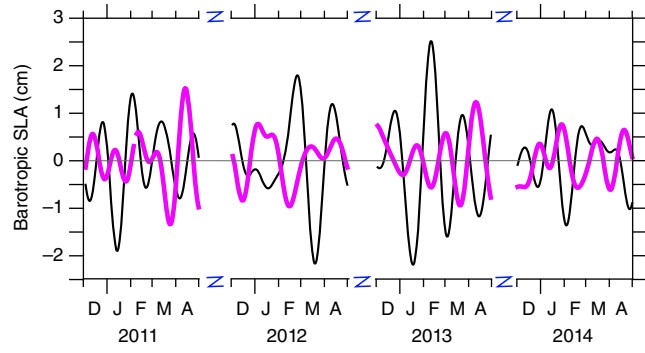

**Fig. 6** Barotropic SLA in the Pacific is out of phase with the barotropic SLA in the BoB. Time series of observed barotropic SLA at BP-BoB (black) and BP-CP (pink, see Fig. 2b for location). $y = 0$ line is plotted for clarity. D, J, F, M, A represent December, January, February, March and April, respectively

## Methods

**BPR data processing**. We have considered 6-year long time series (May 01, 2010–December 31, 2015) from BPRs located at 88.80° E, 6.25° N (BP-BoB1, water depth of 3793 m, NDBC-23227), 88.548° E, 8.809° N (BP-BoB2, water depth of 3472 m, NDBC-23402), BP-AS (65.33° E, 20.80° N, 2612 m, NDBC-23226), BP-NWAB (117.94° E, 15.02° S, 5664 m, NDBC-65003) and BP-CP (125° W, 8.5° S, 4449 m, NDBC-51406). The accuracy of these BPRs is ~1 mm[30]. Few communication gaps in BP-BoB1 and BP-AS were filled using the data recorded on-board (available at Indian National Center for Ocean Information Services). These BPRs measure ocean bottom pressure in pounds per square inch absolute (PSIA). However, the information is disseminated as EWD after applying a constant 670.0 mm of water/PSIA conversion factor. The BPRs have a time resolution of 15 min when operating in the normal mode. However, we chose hourly data by sub-sampling only the 0th minute of every hour from the normal mode data.

During the maintenance of bottom pressure sensor, the sensor is extracted, serviced and subsequently redeployed (typically within few hours) resulting in some changes in the location and depth with respect to its previous deployment. Though the alteration in the location of the sensor during redeployment in the horizontal plane is small (~50 km) compared to the typical scales discussed in the present study, the change in depth from one deployment to another resulted in discontinuities and sharp jumps in the EWD records. Hence, each redeployment yielded a segment of EWD time-series fluctuating about some reference determined by the depth of deployment. To construct a continuous time series of EWD anomaly, averages were removed from each such segment. The anomalies were then subjected to TASK-2000[31] software to remove the tidal frequencies from each segment. The resultant non-tidal data still had few spikes and visible trends either due to the drift of the sensor or due to the sinking of the BPR[30] (Supplementary Figure 1a). The spikes exceeding two standard deviations were then removed. The process of de-tiding and spike-removal was repeated to get a cleaner data set for further analysis. A second-order polynomial was then fitted to each such segment of EWD anomaly to remove the trends (Supplementary Figure 1b). The segments were then concatenated to construct a long time series. There were however still gaps due to communication breaks or due to redeployment. The gaps were typically of 1–2 days except for a single instance when it extended for ~1 month in the BP-NWAB records that occurred in June 2013. All such gaps were subsequently filled using linear interpolation.

After following the aforementioned processing, it was noted that the data from BP-BoB1 after 14 February, 2014 had unreasonable spikes (red curve in Supplementary Figure 1a). Hence, the data from BP-BoB2 from 15 February, 2014 was concatenated with the data from BP-BoB1 during 1 May, 2010 to 14 February, 2014 to obtain a continuous time series extending till 31 December, 2015. The merger of the data from these two BPRs is not expected to introduce errors because the BPRs were located within a distance of 285 km, which is much less than the spatial scale of the physical processes we are interested here. The concatenated time series of BP-BoB1 and BP-BoB2 is henceforth referred to as BP-BoB. Nominally, the location of BP-BoB1 is assigned to BP-BoB.

The bottom pressure data from central Pacific (BP-CP) went through the same data processing mentioned above to remove the occasional presence of spikes even though it was processed by National Oceanic and Atmospheric Administration (NOAA) before uploading it in the web.

**Ocean general circulation model (MOM4p1)**. The OGCM MOM4p1[18] is used to simulate EWD anomaly. The set-up is similar to the one used by ref. [32]. Model domain spans the global ocean between 65° N and 60° S with uniform 0.25° zonal resolution and variable meridional resolution (0.25° at equator and 0.15° at 60° N). It has 40 vertical levels. Bottom topography is based on ETOPO2[33]. The minimum

depth of the ocean is set to 70 m. Shallower regions are deepened to the minimum depth. Unless otherwise mentioned, sea surface temperature (SST) and sea surface salinity (SSS) were relaxed to their daily climatology from NIOA atlas[34]. The vertical mixing scheme is KPP[35] where bulk Richardson number is set at 0.3. For horizontal mixing, a combination of Laplacian and bi-harmonic friction with Smagorinsky mixing coefficient of 0.01 (velocity scale 0.04 m s$^{-1}$) and 0.1 (velocity scale 0.005 m s$^{-1}$) respectively, have been used. The model is forced with 6-hourly National Centre for Medium Range Weather Forecasting (NCMRWF) wind, atmospheric pressure, radiative fluxes, air temperature and specific humidity fields from 2009 to 2016. Precipitation is from NCMRWF. Monthly climatology of the surface chlorophyll concentration inferred from SeaWifs[36] is used to estimate the shortwave radiation penetration depth into the ocean. The baroclinic and barotropic time-steps are set at 800 and 10 s, respectively. Model is spun-up for 100 years from a state of rest forced by the climatological CORE-II[37] forcing and its inter-annual product was used till 2009. Then the forcing is switched to NCMRWF. We consider the 2010–2015 period in the present study.

**Experiments from MOM4p1**. The influence of dynamic atmospheric pressure on barotropic variability in sea-level is known to be an order of magnitude less than that due to wind stress, at intraseasonal time scales[13,38,39]. In order to substantiate this claim, MOM4p1 was forced with and without atmospheric pressure variability (UVP Experiment and UV Experiment, respectively). The experimental settings are listed in Supplementary Table 1. In UV experiment, a constant pressure (1025 hPa) was globally imposed on the ocean surface, keeping the wind as in UVP experiment. The intraseasonal barotropic SLA time series were extracted from both the experiments at the three BPR locations (Supplementary Figure 4). It is seen that the effects of dynamic atmospheric pressure on intraseasonal barotropic SLA are indeed minimal, typically of order or less than 3 mm in absolute values. Henceforth, we do not impose any atmospheric pressure variability on the ocean in all subsequent experiments.

Five sensitivity experiments were carried out with the model by restricting the wind forcing to the boxed region displayed in Fig. 3a. No other fluxes were imposed. Unless otherwise mentioned, SST and SSS were relaxed to their daily climatology. The same bulk parameterisation[40] for wind stress as in UV experiment was used across all the sensitivity experiments. The wind mask was created using a hyperbolic tangent function. To avoid numerical instabilities, the winds were smoothly decayed to 0 at the edges of the box over a length scale of 300 km. The spurious wind stress curl generated by the smoothing at the edges of the box was insignificant (two orders of magnitude less) compared to the wind stress curl inside the box and hence we assume that any aliasing of barotropic SLA due to this spurious curl is minimal and negligible.

**Barotropic bottom pressure**. In order to estimate the barotropic bottom pressure, we followed the methodology of ref. [10] and applied it to our intra-seasonally filtered model simulation. In their work, the anomaly of barotropic bottom pressure is simply defined as the anomaly of the vertically-averaged ocean pressure (see Equation (3) in ref. [10]). We applied this equation to our model outputs (UV Experiment, Supplementary Table 1), and compared it with the model total bottom pressure. Supplementary Figure 2 displays the relative standard difference between these two quantities at intraseasonal time scale. It is seen that, throughout the TIO, the barotropic bottom pressure differs from the total bottom pressure by typically less than 10%. Their maximum difference is seen in the Bay of Bengal, but does not exceed 15%. This implies that, at intra-seasonal timescales, over our domain of interest, the baroclinic bottom pressure can be considered as negligible. As a result, the BPR observations, as well as the model bottom pressure, can be considered as a reliable proxy of barotropic SLA.

**MJO phase diagram**. The state of the MJO (amplitude and phase) is defined using the Real-time Multivariate MJO index (RMM, available online at http://www.bom.gov.au/bmrc/clfor/cfstaff/matw/maproom/RMM/) of ref. [17]. This index defines the MJO through projection of daily anomaly data onto the leading pair of empirical orthogonal functions (EOFs) of the combined fields of equatorially averaged (15° N–15° S) outgoing longwave radiation, 850 hPa zonal wind, and 200 hPa zonal wind. Longer time-scale variability resulting from El Niño-Southern Oscillation (ENSO) and other inter-annual variations with periods longer than about 200 days were removed prior to this projection, but otherwise no temporal filtering is applied. We show a schematic phase diagram of MJO in Supplementary Figure 7. From the two principal component time series (RMM1 and RMM2), a two-dimensional phase space is defined and this is used to define eight distinct phases of MJO (#1 to #8, left $y$ axis in Supplementary Figure 7) with respect to time ($x$ axis in Supplementary Figure 7). Each of these 8 phases is associated with a longitudinal location of the active core of the MJO. This phase diagram denotes the strength of the MJO at each of its phases, viz. at each of its locations along its propagation pathway. The MJO is considered strong when the strength, defined as the amplitude of $\sqrt{RMM1 + RMM2}$, exceeds unity. In the right $y$-axis of Supplementary Figure 7, the typical location of the MJO propagating through the Indo-Pacific basin is indicated, in each of its phases.

**Estimation of the barotropic waves propagation speed**. In order to infer the phase speeds of the various waves identified in the NWAB experiment, we applied a two-dimensional Radon transform to the intraseasonally filtered model barotropic SLA, extracted along the propagation pathways shown in Supplementary Figure 9a. The propagation pathways of the planetary Rossby wave and the topographic Rossby wave were arbitrarily chosen among the various pathways lying westward and north-westward of the NWAB region. Supplementary Figure 9b–d displays the results we obtained: it was found that the planetary Rossby wave propagates at 50 m s$^{-1}$, the topographic Rossby wave propagates at 17 m s$^{-1}$ and the continental shelf wave propagates at 6.5 m s$^{-1}$.

## Data availability

BPR data is available at http://www.ndbc.noaa.gov/dart.shtml, which suffers from missing data for BP-BoB and BP-AS. INCOIS can provide gap-filled BPR data on request for BP-BoB and BP-AS. AVISO data is available at http://www.aviso.altimetry.fr/duacs/. EWD from GRACE was downloaded from https://grace.obs-mip.fr/variable-models-grace-lageos/grace-solutions-release-03/. National Center for Medium Range Weather Forecasting (NCMRWF) fluxes are available from the Indian National Center for Ocean Information Services (INCOIS) on request. Real-time Multivariate MJO index (RMM) is available at http://www.bom.gov.au/bmrc/clfor/cfstaff/matw/maproom/RMM/. North Indian Ocean Atlas (NIOA) is available at http://www.nio.org/index/option/com_nomenu/task/show/tid/2/sid/18/id/229.

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

## Acknowledgements

The lead author is grateful to INCOIS and the Ministry of Earth Sciences (MoES) for providing facilities and the University Grants Commission (UGC) for providing Research Fellowship. The authors thank Ajay Kumar and his team for providing the BPR data and Dr. Abhisek Chatterjee for sharing the MOM4p1 set-up. PLN Murty's help in setting up model experiments is gratefully acknowledged. The authors thank Jean-Michel Lemoine from CNES/GRGS, Toulouse, France for identifying the appropriate GRACE product. The authors thank Damien Allain, Florent Lyard and Loren Carrère for fruitful discussions. The authors thank Dr. Wayne Crawford for improving the manuscript readability. This is INCOIS contribution number 336.

## Author contributions

S.S.C.S., F.D., A.P. and B.R. conceived the idea. B.R. processed the data. B.R., S.P. and M.A. carried out all the control and sensitivity experiments. F.D., L.T., A.P., S.S.C.S. and B.R. analysed the results. A.P., F.D. and B.R. wrote most of the manuscript and S.S.C.S. corrected. All authors contributed to the material of the paper through multiple discussions. S.S.V.S.R. contributed to Fig. 6.

## Additional information

**Competing interests:** The authors declare no competing interests.

