## [Peer Review File · Nature Communications]

Reviewers' comments:

Reviewer #1 (Remarks to the Author):

The authors describe extensive data analysis and modeling work to interpret bottom pressure measurements in the tropical Indian Ocean (TIO) for the period 2010-2015. The results are both interesting and provoking: they highlight the importance of bottom pressure as a novel and complementary view of the tropical ocean intraseasonal variability and provide evidence for importance of winds, stratification and topography in explaining these signals. Main findings in the paper should be an important contribution to the oceanographic literature, but before the paper can be accepted, the presentation needs to be substantially revised to avoid confusing definitions, loose claims and misinterpretation of results, as detailed below.

MAIN POINTS

#1. The implicit assumption made from the start that bottom pressure is equivalent to "barotropic sea level" (BSL) needs to be clarified and correctly framed, and not just taken as a given. Definitions of barotropic and baroclinic sea level and bottom pressure exist in the literature (e.g., Fukumori et al. 1998, Piecuch et al. 2015) and can be adopted here. In particular, Piecuch et al. (2015) include relevant discussions on the relation between bottom pressure and sea level and how bottom pressure variability can be related to both barotropic and baroclinic dynamics, especially in the tropics. Piecuch (2015) provides an example of bottom pressure signals associated with low-latitude baroclinic Rossby waves. The large scale nature of the intraseasonal signals discussed in the present work is likely to mostly represent barotropic processes, but such inference could be tested, for example, using the modeling results, along some of the methods in Piecuch et al. (2015). Many statements related to the "barotropic" assumption also need to be corrected throughout the paper (e.g., lines 51-53, 58).

Fukumori, I., R. Raghunath, and L.-L. Fu, 1998: Nature of global large-scale sea level variability in relation to atmospheric forcing: A modeling study. *J. Geophys. Res.*, 103, 5493–5512, doi:10.1029/97JC02907.

Piecuch, C. G., 2015: Bottom-pressure signature of annual baroclinic Rossby waves in the northeast tropical Pacific Ocean, *J. Geophys. Res. Oceans*, 120, 2449–2459, doi:10.1002/2014JC010667.

Piecuch, C. G., I. Fukumori, R. M. Ponte, and O. Wang, 2015: Vertical structure of ocean pressure fluctuations with application to satellite-gravimetric observations, *J. Atmos. Oce. Tech.*, 32, 603–613.

#2. The paper pretends to highlight the importance of barotropic dynamics to explain intraseasonal sea level variability in the TIO, but the quantitative claims in lines 18-20, 95-97 need to be better explained. Before these claims can have any meaning, the authors need to state clearly how results are calculated, in terms of ratios of either standard deviation or variance of bottom pressure and sea level (e.g., from altimetry), and what filtering in space/time (if any) is applied to the time series.

#3. The importance of the winds in the box in Fig 3a is overemphasized and needs to be discussed in a more balanced way. The large amplitude differences in Fig 4a between UV (red) and NWAB (green) experiments indicate that winds outside the outlined region in Fig 3a are important for the variability. The correlation in Fig 3a is also not impressive: significance is provided at 90% level and I am assuming that at 95% level, regions of significant correlation would be mostly gone. There is also lack of coherence between the BP-AS record in the Arabian Sea and the others, suggesting that the large scale signal is corrupted by other shorter scale or more local variability. A comment on the disparate behavior of the BP-AS record also seems warranted.

#4. The discussion of phase speeds (lines 229-247) needs to be clarified. Visual inference of phase speeds from Fig 4 suppl is only convincing for the (faster) continental shelf wave. A range of phase speeds is quoted in the text and Fig 4 suppl, but it is not clear how those numbers are derived. There are, of course, quantitative methods such as Radon transforms for deriving phase speeds from diagrams such as those in Fig 4 suppl, if the authors want to be quantitative about it.

#5. I find many potential issues with the text in the "Discussion and Implications" section. The use of "distinctly" in line 255 seems to be an overstatement. While out-of-phase behavior in Fig 6 is clear for 2013, the same cannot be said for 2011 and part of the records in 2012, 2014. The "prevalence" of large barotropic variability (lines 264-265) has not been demonstrated sufficiently in the results shown (see #2 above). Regarding points in lines 265-269, intraseasonal signals at time scales of 30 days and longer, as discussed here, are not supposed to be removed from satellite measurements unless they are aliased or static in nature. Suggestion that the variability discussed in this paper needs to be removed or corrected is uncalled for. Statement on lines 271-272 needs to be clarified as already discussed (see again #2 above). The reference to "inverse cascade" (line 278) in this context can be confusing: it normally refers to high wavenumber, turbulent regimes. The discussion at the end suggesting the importance of present results to understand sea level trends (lines 289-294) sounds somewhat farfetched. What do the authors mean by "improper accounting for IBSLA variability in a study targeting decadal or longer time scales"?

#6. The citations are in several cases inappropriate and hardly justified. For example, I do not think that ref. 8 should be cited in line 53 (after all that statement is incorrect as per #1 above). In lines 103 and 273, Vivier et al. and Fu and Davidson citations, respectively, are appropriate for mid and high latitudes, not the tropics. In line 107, statement and citation of Rietbroek et al. is inappropriate. Discussion of intraseasonal signals does not call for citing Leuliette and Willis on sea level budget for much longer time scales (line 288). The authors need to cite more judiciously in a revised manuscript.

OTHER COMMENTS AND SUGGESTED EDITS (BY LINE #)

2. "...revealed by bottom pressure"

abstract. As written, there is no mention of bottom pressure, only "barotropic sea level". This is not what is examined directly in this paper. The abstract should reflect more clearly what is analyzed and what is assumed.

23. "...fluctuations are likely..."

26-28. Statement is vague and does not add anything useful. I don't think it belongs in the abstract.

40. "quasi-geostrophy"

42-45. Delete repetitive text at end of sentence: "...whereas it is ...components"

47-49. Clarify that the large scale nature of the MJO is the key ingredient here.

55-56. Remove redundant text: "The BPRs data can be expressed..."

Fig 1. Describe in caption what the x-axis labels "A" and "D" refer to in panels b,c,d.

61-64. Statement is confusing and ultimately unnecessary. I suggest deleting it. Any clarification of the role of atmospheric pressure forcing can be provided later, as discussed in model results.

77-78. Apart from NWAB record (Fig 1d), it is hard to see the claimed enhanced variability during DJFMA in BoB and AS records (Fig 1b,c). Can this statement be better quantified by providing, for example, the ratio of average variance in DJFMA to that in the remaining months?

98. This statement does not seem entirely justified. My view is that BP-AS is visually quite different.

121. Strictly speaking you probably want the forcing to be defined as a scalar, not as a vector.

Fig 3. Clarify in the text or caption if the correlation is based on band-passed or raw time series.

148. State which model solution is referred to here.

149-151. Clarify in text if modeled IBSLA in Fig 4 is just taken to be the bottom pressure in the model.

184-187. Statement is hard to follow, particularly the corollary statement in lines 186-188.

206-220. I had a hard time relating the text about the forcing "patches" (206-212) to what is seen in the movie. In any case, it is not clear what the relevance of these "patches" is to what follows. The use of "downwelling/upwelling terminology is more appropriate for baroclinic waves, not for barotropic waves discussed in this context. Perhaps more appropriate to talk about waves associated with positive/negative sea level or bottom pressure anomalies. It is also very hard to see most of the wave propagation claimed in the movie, which is not surprising because the phase speeds seem to be too fast and the waves can cross the basin in a couple of days.

234-236. Judging from the movie, apart from killing the topographic waves, the flat bottom experiment seems to have no Indian basin-scale response either. Why? Can the authors comment on whether the main effect of topography is through the direct modification of the vorticity input (through the curl τ/H term) or through other more complex dynamical effects (barotropic/baroclinic coupling, JEBAR)?

283. "...to realistically simulate..."

Fig 4 suppl. Caption should state what is being plotted. IBSLA? Need also units and color bar.

Reviewer #2 (Remarks to the Author):

This paper demonstrates sea level variation, which is horizontally coherent over the tropical Indian Ocean at the intraseasonal time-scale, and its possible mechanism using bottom pressure records in three locations and satellite data, together with a series of sensitivity experiments of an OGCM. The authors suggest that the intraseasonal sea level signals appeared in the data are caused by barotropic ocean responses to the atmospheric wind variations associated with the MJO. It has been widely believed that the sea level variations due to barotropic ocean responses in the tropical regions would be small compared to the one associated with the baroclinic responses, and the signals are mostly neglected from previous analyses. The authors of this paper try to investigate in detail this overlooked component of the ocean responses and clearly demonstrate that about 30% of the intraseasonal sea level signals could be due to this barotropic response. This paper is, therefore, worth publishing in terms of the topic covered by the paper. However, the analyses conducted and particularly the arguments on dynamics of the ocean responses are insufficient and includes some speculations, as listed below. It would be recommended to revise the paper to include more clarifications on processes involved, so that the revised manuscript will strengthen the arguments on the main conclusions of the paper.

Major points

(1) The authors pointed out that the atmospheric disturbances that generate the oceanic barotropic responses are the MJOs. Considering the proposed forcing region and seasons of the year, the MJO related disturbances may be the most possible candidate. However, in Fig.3(a), the correlation between the BP-BoB and wind forcing only indicates negative values. The wind stress curl associated with the MJO type zonal wind disturbances usually show a dipole structure in meridional in the curl field. Is this Fig.3(a)-distribution associated with the bottom topography? It would be nice if we can see how the MJO type zonal winds generate such monopole forcing structure.

(2) Fig.4 clearly indicates that the BP-BoB signals can be generated by the winds over the NWAB region. However, the amplitude of the NWAB sensitivity experiment (green curve) is about a half of the UV experiment (red curve). This suggests only about a half the amplitude is explained by the winds in the NWAB region. Why the amplitude is weak in this case? What are the other possible forcing candidates?

(3) The discussions of the dynamics of barotropic wave propagation is interesting. However, it is not yet clear by the sensitivity experiments about how the proposed three types of waves affect the BoB and other regions. For example, the NWAB-FB case exclude the shelf waves and topographic Rossby waves, but at the same time, the forcing in the NWAB region by the wind stress are significantly reduced due to the constant depth. This leads significant amplitude reduction for the planetary Rossby wave component as well, and it is not clear from this sensitivity case that the former two waves are also important.

The authors suggest that this planetary Rossby wave propagate northwestward by showing distance-time section in Supplementary Fig.4(b), but determination of the phase speed from the figure is too subjective and includes significant error due to a coarse resolution in time (at least from the figure). The signals in the X-T diagram are series of isolated maxima and minima, which may not indicate smooth propagation but, instead, a superposition of several different signals. What determines this direction of the wave propagation? The Ninety-East Ridge is aligned almost north-south direction and there are other similar ridges in the tropical Indian Ocean sector. Why only the Ninety-East Ridge is highlighted? The basin-scale adjustment of the barotropic waves should be shown clearly, e.g. differences between the cases with/without the bottom topography outside the forcing region.

Also, the topographic Rossby wave pathway seems to be selected subjectively. It may be possible that the topographic Rossby wave propagates westward, but why this wave is significant along a particular f/H values? How this topographic Rossby wave generated in the region off west coast of Australia and how the wave affects the BP-BoB signals?

The authors neglected all these (and many other) relevant questions for the dynamics part of the paper, which is a weak point of this manuscript. This part should be revised with a carefully arranged set of the sensitivity experiments.

Reviewer #3 (Remarks to the Author):

The manuscript describes a fluctuation of ocean bottom pressure found in the tropical Indian Ocean based on observations and a numerical ocean circulation model. The study concludes that this intra-seasonal fluctuation corresponds to waves driven by winds in the north-west Australian basin associated with the Madden-Julian Oscillation (MJO). Importantly, the paper argues that the fluctuation is barotropic and accounts for far larger variations in sea level ("about 30%") than commonly thought, calling into question the validity of other studies concerning the sea level budget.

The manuscript's tenet that bottom pressure recorders "record variabilities attributed to barotropic

processes only" is incorrect. (There are no page numbers or line numbers in the document, rendering references to sections of the manuscript cumbersome.) Pressure in the ocean is in hydrostatic balance and its variation at depth corresponds to the vertical integral of anomalies above it that include both barotropic and baroclinic processes. In particular, baroclinic pressure variations are not necessarily negligible at the bottom.

A number of other assertions made in the manuscript are also questionable. The study focuses on variations between 30- to 80-day periods, which is but a small part of sea level (and ocean bottom pressure) fluctuation. Sea level budgets can hardly be assessed from such a limited study. Moreover, the paper's conclusion that bottom pressure accounts for 30% of the sea level variance is not demonstrated. In fact, the correlation between ocean bottom pressure and sea level being small in both amplitude and spatial extent, shown in Figure 2, suggests that the former's contribution to the latter, even if it were equivalent to barotropic sea level, is secondary. The correlation's spatial extent does not show the fluctuation being "in unison" across the tropical Indian Ocean either.

In summary, the manuscript is based on false assumptions and questionable claims (e.g., that bottom pressure corresponds to barotropic variation mentioned above). Conclusions are not supported by the evidence provided and are misleading (see also other examples below). The focus of the paper (variations within periods of 30 to 80 days) is also too narrow to be of interest to the oceanographic audience of Nature Communications.

I cannot recommend publication of this manuscript.

Other comments:

- 1) Description of "inverse barometric response" is incorrect. Inverse barometer response is isostatic, meaning changes in atmospheric pressure is compensated by changes in sea level. By definition, there is no bottom pressure variation associated with an inverse barometer response.
- 2) Use of numerous unconventional abbreviations (e.g., BSL, TIO, DJFMA, IBSLA, PSIA, EWD) obfuscate the discussion.
- 3) The discussion on spatial scales and their significance are questionable. There are plenty of intra-seasonal variations that exceed 100km in horizontal scale that are not "trapped near the ocean surface" (e.g., equatorial Kelvin waves). Horizontal scales do not inform vertical modes either. For instance, baroclinic variations (e.g., Indian Ocean Dipole Mode) as well as barotropic changes can have scales exceeding 1000km, and hence to say that "coarse-grained SLA results from barotropic processes" is incorrect.
- 4) The relationship between wind stress and the Madden-Julian Oscillation (and therefore the significance of MJO to the subject ocean bottom pressure fluctuation) is unclear. For instance, it is not shown how much of the wind stress variability is actually due to the Madden-Julian Oscillation.
- 5) The model correlation (Figure 4) has much larger amplitude with wider spatial extent than do observations (Figure 2). The model does not "reproduce" observations as claimed.
- 6) The model experiment with spatially confined forcing (experiment NWAB) has less than half the amplitude of the one with full forcing (UV) (Figure 4), which suggests that most of the relevant forcing exist elsewhere. Therefore, the conclusion that the variations observed at the bottom pressure recorders are driven by "wind stress over the Eastern Indian Ocean" is suspect.
- 7) Ocean circulation is continuous and, as such, a "quantum" of flux makes no sense. The waves referred to in the manuscript are neither demonstrated in the study nor are they evident in the animation ("video") provided as supplemental material. A barotropic planetary Rossby wave that travels 45 m/s would have a period of no more than 10-days, not the 30 to 80 days found in the observations. The animation shows the Indian Ocean changing its net mass without compensating changes elsewhere within the plotted domain. It is argued that the mass exchange occurs remotely with the central Pacific Ocean (Figure 6), but the explained waves originate in the Indian Ocean and not the Pacific. There is a logical disconnection in the explanation.
- 8) Removing intra-seasonal barotropic variation is not a "standard" approach in satellite data

processing. One often does use "2D barotropic models" when utilizing satellite data, but only to remove high frequency variations that can otherwise be aliased. The Nyquist frequency for the constellation of satellite sea level measurements in operation today is less than 10-days, which is far shorter than the 30 to 80-day period analyzed in this study. Effects of atmospheric pressure loading that deviate from inverse barometer response is generally considered to be shorter than a few days, again much shorter than the period analyzed here. None of these approaches "miss a large fraction of the actual signal."

9) Barotropic Rossby waves was said to have a speed of 45 m/s, which takes 2-days to cross the Indian Ocean. This cannot possibly adjust the Indian Ocean in a "few hours" as stated in "Discussion and Implications". The MJO response discussed earlier was also in terms of barotropic waves, not baroclinic variability.

Reviewer #1

Main Points

#1. *The implicit assumption made from the start that bottom pressure is equivalent to “barotropic sea level” (BSL) needs to be clarified and correctly framed, and not just taken as a given. Definitions of barotropic and baroclinic sea level and bottom pressure exist in the literature (e.g., Fukumori et al. 1998, Piecuch et al. 2015) and can be adopted here. In particular, Piecuch et al. (2015) include relevant discussions on the relation between bottom pressure and sea level and how bottom pressure variability can be related to both barotropic and baroclinic dynamics, especially in the tropics. Piecuch (2015) provides an example of bottom pressure signals associated with low-latitude baroclinic Rossby waves. The large scale nature of the intraseasonal signals discussed in the present work is likely to mostly represent barotropic processes, but such inference could be tested, for example, using the modeling results, along some of the methods in Piecuch et al. (2015). Many statements related to the “barotropic” assumption also need to be corrected throughout the paper (e.g., lines 51-53, 58).*

Fukumori, I., R. Raghunath, and L.-L. Fu, 1998: Nature of global large-scale sea level variability in relation to atmospheric forcing: A modeling study. J. Geophys. Res., 103, 5493–5512, doi:10.1029/97JC02907.

Piecuch, C. G., 2015: Bottom-pressure signature of annual baroclinic Rossby waves in the northeast tropical Pacific Ocean, J. Geophys. Res. Oceans, 120, 2449–2459, doi:10.1002/2014JC010667.

Piecuch, C. G., I. Fukumori, R. M. Ponte, and O. Wang, 2015: Vertical structure of ocean pressure fluctuations with application to satellite-gravimetric observations, J. Atmos. Oce. Tech., 32, 603–613.

Reply: We thank the Reviewer for pointing out that baroclinic processes can also influence bottom pressure anomalies. We have now re-examined our results. We have followed the prescription of Piecuch et al (Piecuch, C. G., I. Fukumori, R. M. Ponte, and O. Wang, 2015: *Vertical structure of ocean pressure fluctuations with application to satellite-gravimetric observations, J. Atmos. Oce. Tech., 32, 603–613*) and diagnosed the relative influence of the barotropic processes on the total bottom pressure simulated by our numerical model. Our results show that, even though bottom pressure can be modulated by baroclinic processes, the bottom pressure anomaly and the barotropic sea level anomaly (SLA) are equivalent over our domain of interest at intraseasonal time scales, to a very large extent. Indeed, we prove that the relative effect of baroclinicity in bottom pressure anomalies is typically less than 10% in the tropical Indian Ocean at intraseasonal time scales. This is explained explicitly in the revised manuscript (lines 50-53; 108-111) and discussed in a full new section (section 3) of the revised Methods, as well as with a new figure (Supplementary Fig. 3).

#2 *The paper pretends to highlight the importance of barotropic dynamics to explain intraseasonal sea level variability in the TIO, but the quantitative claims in lines 18-20, 95-97 need to be better explained. Before these claims can have any meaning, the authors need to state clearly how results are*

calculated, in terms of ratios of either standard deviation or variance of bottom pressure and sea level (e.g., from altimetry), and what filtering in space/time (if any) is applied to the time series

Reply: We agree. We now present the ratio of standard deviation of intraseasonal barotropic SLA and the total intraseasonal SLA derived from altimetry at the three bottom pressure recorder locations in the revised manuscript (lines 114-117). This fraction is quite large (30% - 60%) during December-April. We have also illustrated the time series of intraseasonal barotropic SLA and the total intraseasonal SLA from altimetry in a new figure (Supplementary Fig. 4) to provide a visual experience to the readers about the relative contribution of barotropic SLA at intraseasonal scales. Regarding the filtering applied to the time series, our analysis is devoted to the intraseasonal band, as mentioned earlier in the text (lines 102 - 103). Regarding the filtering in space, nothing was done, so we do not mention anything in the revised manuscript.

#3 The importance of the winds in the box in Fig 3a is overemphasized and needs to be discussed in a more balanced way. The large amplitude differences in Fig 4a between UV (red) and NWAB (green) experiments indicate that winds outside the outlined region in Fig 3a are important for the variability. The correlation in Fig 3a is also not impressive: significance is provided at 90% level and I am assuming that at 95% level, regions of significant correlation would be mostly gone. There is also lack of coherence between the BP-AS record in the Arabian Sea and the others, suggesting that the large scale signal is corrupted by other shorter scale or more local variability. A comment on the disparate behavior of the BP-AS record also seems warranted.

Reply: We agree. We have revised thoroughly our results concerning NWAB experiment and other sensitivity experiments. It turns out that the model forcing strategy we had used for the sensitivity experiments (including NWAB experiment) in the previous version was corrupted by an erroneous formulation of the wind-speed - to - wind-stress conversion in the CORE bulk formulae, yielding an underestimation of the wind stress applied to the model by a factor of ~ 2 , compared to our UV reference experiment (which was correct, and has not been changed in the revised version). This mistake happened because we had computed off-line the wind stress (from wind speed) for all the regional sensitivity experiments, based on a too low non-dimensional drag coefficient ($C_d=0.0012$), inconsistent with the CORE bulk formulation embedded in the model and used on-line for the UV experiment. We have now used the same CORE bulk formulation throughout all the experiments, including the regionally-forced sensitivity experiments. This bulk formulation increases the wind stress by a factor of ~ 2 in all sensitivity experiments compared to the earlier version. Consequently, the amplitude of barotropic SLA in the NWAB experiment has also considerably increased. The NWAB experiment now reproduces the UV

experiment (and hence the observations) to a large extent, particularly during strong MJO events (see the green and red curves in the new Fig. 4b of the revised manuscript). Therefore, we conclude that the NWAB region appears as the primary forcing region for the barotropic fluctuations observed in the tropical Indian Ocean, suggesting that the contribution of the forcing from the other regions is relatively small, particularly during strong MJO events. That is now specifically mentioned in the Methods (subsection “Experiments from MOM4p1”). We believe that the new results are much more convincing than in the earlier version, and that they address the concern of the Reviewer.

We agree with the Reviewer that the forcing region considerably shrinks if we admit only 95% significance in the correlation. However, we have decided to stick to 90% significance in order to make physical sense of the geographical region highlighted by the correlation pattern.

Regarding the disparate behaviour of BP-AS, we included a specific comment in the revised manuscript (lines 98-101). We believe that BP-AS, because of its location close to the continental slope, is relatively susceptible to baroclinic processes.

#4. The discussion of phase speeds (lines 229-247) needs to be clarified. Visual inference of phase speeds from Fig 4 suppl is only convincing for the (faster) continental shelf wave. A range of phase speeds is quoted in the text and Fig 4 suppl, but it is not clear how those numbers are derived. There are, of course, quantitative methods such as Radon transforms for deriving phase speeds from diagrams such as those in Fig 4 suppl, if the authors want to be quantitative about it.

Reply: We have followed the Reviewer's advice and we now rely on 2-D Radon transform to objectively infer the phase speeds of the waves we analyze. This is included as a new section in Methods (section 5) and the corresponding Supplementary Fig. 8.

#5. I find many potential issues with the text in the “Discussion and Implications” section. The use of “distinctly” in line 255 seems to be an overstatement. While out-of-phase behavior in Fig 6 is clear for 2013, the same cannot be said for 2011 and part of the records in 2012, 2014. The “prevalence” of large barotropic variability (lines 264-265) has not been demonstrated sufficiently in the results shown (see #2 above). Regarding points in lines 265-269, intraseasonal signals at time scales of 30 days and longer, as discussed here, are not supposed to be removed from satellite measurements unless they are aliased or static in nature. Suggestion that the variability discussed in this paper needs to be removed or corrected is uncalled for. Statement on lines 271-272 needs to be clarified as already discussed (see again #2 above). The reference to “inverse cascade” (line 278) in this context can be confusing: it normally refers to high wavenumber, turbulent regimes. The discussion at the end suggesting the importance of present results to understand sea level trends (lines 289-294) sounds

somewhat farfetched. What do the authors mean by “improper accounting for IBSLA variability in a study targeting decadal or longer time scales”?

Reply: We agree. We have completely re-written this section, to make it more focussed and more relevant (including removal of our earlier points on sea level trends and inverse cascading of energy). We have now shown that the intraseasonal barotropic response of the tropical Indian Ocean is indeed significant as it explains 30% - 60% of the total intraseasonal SLA variability. We have also taken care to avoid the overemphasis on the implication of these intraseasonal barotropic fluctuations.

#6. The citations are in several cases inappropriate and hardly justified. For example, I do not think that ref. 8 should be cited in line 53 (after all that statement is incorrect as per #1 above). In lines 103 and 273, Vivier et al. and Fu and Davidson citations, respectively, are appropriate for mid and high latitudes, not the tropics. In line 107, statement and citation of Rietbroek et al. is inappropriate. Discussion of intraseasonal signals does not call for citing Leuliette and Willis on sea level budget for much longer time scales (line 288). The authors need to cite more judiciously in a revised manuscript.

Reply: We have carefully cleaned our citations and references list, and replaced all inappropriate citations the Reviewer had pointed out by relevant ones.

Minor points

2. “...revealed by bottom pressure”

abstract. As written, there is no mention of bottom pressure, only “barotropic sea level”. This is not what is examined directly in this paper. The abstract should reflect more clearly what is analyzed and what is assumed.

Reply: This is addressed in the revised abstract (line 17).

23. “...fluctuations are likely...”

Reply: Corrected.

26-28. *Statement is vague and does not add anything useful. I don't think it belongs in the abstract.*

Reply: We removed this statement.

40. *“quasi-geostrophy”*

Reply: Corrected.

42-45. *Delete repetitive text at end of sentence: “...whereas it is ...components”*

Reply: We deleted this.

47-49. *Clarify that the large scale nature of the MJO is the key ingredient here.*

Reply: We begin by the hypothesis that large scale MJOs can potentially trigger barotropic waves (lines 44-47). We believe that the results presented in the manuscript convincingly show that this hypothesis is largely verified in the tropical Indian Ocean.

55-56. *Remove redundant text: “The BPRs data can be expressed...”*

Reply: Corrected.

Fig 1. Describe in caption what the x-axis labels “A” and “D” refer to in panels b,c,d.

Reply: We added the description in Fig. 1b-d.

61-64. *Statement is confusing and ultimately unnecessary. I suggest deleting it. Any clarification of the role of atmospheric pressure forcing can be provided later, as discussed in model results.*

Reply: We agree and deleted this statement.

77-78. *Apart from NWAB record (Fig 1d), it is hard to see the claimed enhanced variability during DJFMA in BoB and AS records (Fig 1b,c). Can this statement be better quantified by providing, for example, the ratio of average variance in DJFMA to that in the remaining months?*

Reply: We agree. We have addressed this concern in the revised manuscript (lines 96-98). We show that the increase in standard deviation in intraseasonal barotropic SLA during December-April with respect to the rest of the months is largest at BP-NWAB (57 %) and smallest at BP-BoB (15 %).

98. *This statement does not seem entirely justified. My view is that BP-AS is visually quite different.*

Reply: We agree. We have re-formulated this statement to explicitly point towards the distinct behaviour of BP-AS record., for clarity (line 98 - 101).

121. Strictly speaking you probably want the forcing to be defined as a scalar, not as a vector.

Reply: Corrected. We now specifically mention that as the vertical component of the forcing (line 130-132).

Fig 3. Clarify in the text or caption if the correlation is based on band-passed or raw time series.

Reply: This has been clarified in the caption.

148. State which model solution is referred to here.

Reply: Addressed (see line 165).

149-151. Clarify in text if modeled IBSLA in Fig 4 is just taken to be the bottom pressure in the model.

Reply: We have clarified in the revised manuscript that the time series is of intraseasonal barotropic SLA derived from the model (lines 162-163). Fig. 4 is revised.

184-187. Statement is hard to follow, particularly the corollary statement in lines 186-188.

Reply: We agree. For clarity, we have removed Fig. 4b (from the earlier version of the manuscript) and the associated discussion in the revised manuscript.

206-220. I had a hard time relating the text about the forcing “patches” (206-212) to what is seen in the movie. In any case, it is not clear what the relevance of these “patches” is to what follows. The use of “downwelling/upwelling terminology is more appropriate for baroclinic waves, not for barotropic waves discussed in this context. Perhaps more appropriate to talk about waves associated with positive/negative sea level or bottom pressure anomalies. It is also very hard to see most of the wave propagation claimed in the movie, which is not surprising because the phase speeds seem to be too fast and the waves can cross the basin in a couple of days.

Reply: We agree. We have omitted the relevance of the forcing patches in the revised manuscript. Instead of downwelling/upwelling, we have followed the advice of the Reviewer and changed it to positive/negative sea level anomalies. Regarding the limited visibility of very fast barotropic transients, we believe the objective evaluation of the phase speeds of the waves (provided in the revised manuscript) bring more confidence to the readers.

234-236. Judging from the movie, apart from killing the topographic waves, the flat bottom experiment seems to have no Indian basin-scale response either. Why? Can the authors comment on

whether the main effect of topography is through the direct modification of the vorticity input (through the curl tau/H term) or through other more complex dynamical effects (barotropic/baroclinic coupling, JEBAR)?

Reply: We believe that it is not correct to assume that there is no basin scale response in the Indian Ocean. We have conducted a series of new sensitivity experiments by varying the depth in the NWAB basin, which show that the basin-wide response can be large if the depth H is small (as the vorticity input to the ocean due to the winds is large at smaller depths). We have illustrated this in Supplementary Fig. 6 of the revised version.

283. "...to realistically simulate..."

Reply: Corrected.

Fig 4 suppl. Caption should state what is being plotted. IBSLA? Need also units and color bar.

Reply: Corrected. This is now Fig. 8 in the revised Supplementary Figures.

Reviewer #2

Main Points

(1) The authors pointed out that the atmospheric disturbances that generate the oceanic barotropic responses are the MJOs. Considering the proposed forcing region and seasons of the year, the MJO related disturbances may be the most possible candidate. However, in Fig.3(a), the correlation between the BP-BoB and wind forcing only indicates negative values. The wind stress curl associated with the MJO type zonal wind disturbances usually show a dipole structure in meridional in the curl field. Is this Fig.3(a)-distribution associated with the bottom topography? It would be nice if we can see how the MJO type zonal winds generate such monopole forcing structure.

Reply: The wind stress curl associated to MJO does show a meridional dipole structure over the north-west Australian Basin (as the Reviewer rightly points). The barotropic ocean, however, responds to $f (\nabla \times \vec{\tau}/H)$ where f is the Coriolis parameter, $\vec{\tau}$ is the wind stress and H is the ocean depth (see the revised manuscript, line 129). It turns out that the steep gradients of ocean topography over the boxed region (north-west Australian Basin) considerably distort the MJO-related dipole of wind stress resulting in the monopole of correlation as seen in Fig. 3 in the manuscript. We have incorporated an explanation in the revised manuscript to make it clear to the readers (lines 141-143).

(2) Fig.4 clearly indicates that the BP-BoB signals can be generated by the winds over the NWAB region. However, the amplitude of the NWAB sensitivity experiment (green curve) is about a half of the UV experiment (red curve). This suggests only about a half the amplitude is explained by the winds in the NWAB region. Why the amplitude is weak in this case? What are the other possible forcing candidates?

Reply: We agree. This comment is in line with Point#3 of the first Reviewer and Point#6 of the third Reviewer. We have revised thoroughly our results concerning NWAB experiment and other sensitivity experiments. It turns out that the model forcing strategy we had used for the sensitivity experiments (including NWAB experiment) in the previous version was corrupted by an erroneous formulation of the wind-speed - to - wind-stress conversion in the CORE bulk formulae, yielding an underestimation of the wind stress applied to the model by a factor of ~ 2 , compared to our UV reference experiment (which was correct, and has not been changed in the revised version). This mistake happened because we had computed off-line the wind stress (from wind speed) for all the regional sensitivity experiments, based on a too low non-dimensional drag coefficient ($C_d=0.0012$), inconsistent with the CORE bulk formulation embedded in the model and used on-line for the UV experiment. We have now used the same CORE bulk formulation throughout all experiments, including the regionally-forced sensitivity experiments. This bulk formulation increases the wind stress by a factor of ~ 2 in all sensitivity experiments compared to the earlier version. Consequently, the amplitude of barotropic SLA in the NWAB experiment has also considerably increased. The NWAB experiment now reproduces the UV experiment (and hence the observations) to a large extent, particularly during strong MJO events (see the green and red curves in the new Fig. 4b of the revised manuscript). Therefore, we conclude that the NWAB region appears as the primary forcing region for the barotropic fluctuations observed in the tropical Indian Ocean, suggesting that the contribution of the forcing from the other regions is relatively small, particularly during strong MJO events. That is now specifically mentioned in the Methods (subsection "Experiments from MOM4p1"). We believe that the new results are much more convincing than in the earlier version, and that they address the concern of the Reviewer.

(3) The discussions of the dynamics of barotropic wave propagation is interesting. However, it is not yet clear by the sensitivity experiments about how the proposed three types of waves affect the BoB and other regions. For example, the NWAB-FB case exclude the shelf waves and topographic Rossbe waves, but at the same time, the forcing in the NWAB region by the wind stress are significantly reduced due to the constant depth. This leads significant amplitude reduction for the planetary Rossby wave component as well, and it is not clear from this sensitivity case that the former two waves are also important.

Reply: We agree. We have thoroughly revised this section to address this point, and included additional sensitivity experiments by varying the depth in the NWAB-FB experiment. They show that the amplitude of the response can indeed significantly vary depending on the chosen depth (Supplementary Fig. 6). This is now discussed in the revised manuscript (lines 196 – 199; 239-248).

The authors suggest that this planetary Rossby wave propagate northwestward by showing distance-time section in Supplementary Fig.4(b), but determination of the phase speed from the figure is too subjective and includes significant error due to a coarse resolution in time (at least from the figure). The signals in the X-T diagram are series of isolated maxima and minima, which may not indicate smooth propagation but, instead, a superposition of several different signals.

Reply: We have now used 2D Radon Transform to objectively infer the phase speeds of the waves (section 5 in revised Methods; Supplementary Fig. 8).

What determines this direction of the wave propagation? The Ninety-East Ridge is aligned almost north-south direction and there are other similar ridges in the tropical Indian Ocean sector. Why only the Ninety-East Ridge is highlighted? The basin-scale adjustment of the barotropic waves should be shown clearly, e.g. differences between the cases with/without the bottom topography outside the forcing region.

Reply: We agree. We have thoroughly revised this section, and included an additional sensitivity experiment (NWAB-NR; Supplementary Table 1; see the additional video 4 provided as supplementary material) to decipher the specific role of the topographic features (outside the NWAB region) in the propagation of the barotropic waves that we identified (lines 239-248). The removal of all ridges outside of NWAB region (NWAB-NR experiment) shows that the topography outside the NWAB region modulates the direction of propagation of these waves. The Ninety-East Ridge particularly favours the north-westward propagation of the planetary Rossby waves, but does not completely precludes westward propagation. This is evident from the differences in the response of the ocean to wind stress over NWAB region (see video 1 and video 4).

Also, the topographic Rossby wave pathway seems to be selected subjectively. It may be possible that the topographic Rossby wave propagates westward, but why this wave is significant along a particular f/H values? How this topographic Rossby wave generated in the region off west coast of Australia and how the wave affects the BP-BoB signals?

The authors neglected all these (and many other) relevant questions for the dynamics part of the paper, which is a weak point of this manuscript. This part should be revised with a carefully arranged set of the sensitivity experiments.

Reply: Video 1 shows that the waves emanate from the NWAB basin and propagate both in the westward and north-westward direction. The phase speeds were determined along an arbitrarily chosen path (mentioned in the revised Methods; section 5). We are not specifying any particular f/H contour in the revised manuscript anymore. Instead we state that the topographic Rossby waves move along f/H contours. We have addressed the specific role played by the Ninety East Ridge as well as by the Southeast Indian Ridge in the revised manuscript (lines 239 - 248). We have shown that the Southeast Indian Ridge acts as a barrier to the barotropic Rossby waves and shields the southern regions from its effects. Removing this ridge (NWAB-NR experiment; video 4) allows the waves to leak to the southern parts (and hence to find a pathway to export energy to the Southern Ocean). This thereby reduces the amplitude of barotropic SLA at BP-BoB by $\sim 30\%$.

Reviewer #3

Main points

The manuscript's tenet that bottom pressure recorders "record variabilities attributed to barotropic processes only" is incorrect. (There are no page numbers or line numbers in the document, rendering references to sections of the manuscript cumbersome.) Pressure in the ocean is in hydrostatic balance and its variation at depth corresponds to the vertical integral of anomalies above it that include both barotropic and baroclinic processes. In particular, baroclinic pressure variations are not necessarily negligible at the bottom.

Reply: We thank the Reviewer for pointing out that baroclinic processes can also influence bottom pressure anomalies. We have now re-examined our results. We followed the prescription of Piecuch et al (*Piecuch, C. G., I. Fukumori, R. M. Ponte, and O. Wang, 2015: Vertical structure of ocean pressure fluctuations with application to satellite-gravimetric observations, J. Atmos. Oce. Tech., 32, 603–613*) and diagnosed the relative influence of the barotropic processes on the total bottom pressure simulated by our numerical model. Our results show that, even though bottom pressure can be modulated by baroclinic processes, the bottom pressure anomaly and the barotropic sea level anomaly (SLA) are equivalent over our domain of interest at intraseasonal time scales, to a very large extent. Indeed, we prove that the relative effect of baroclinicity in bottom pressure anomalies is typically less than 10% in the tropical Indian Ocean at intraseasonal time scales. This is explained explicitly in the revised manuscript (lines 50-53; 108-111) and discussed in a full new section (section 3) of the revised Methods, as well as with a new figure (Supplementary Fig. 3).

In the revised manuscript we acknowledge that baroclinic processes can also modulate the bottom pressure anomalies (lines 50 - 53). Regarding absence of page numbers, this is

puzzling as the other reviewers visibly had access to the pages and line numbers. We are not sure why that is so.

A number of other assertions made in the manuscript are also questionable. The study focuses on variations between 30- to 80-day periods, which is but a small part of sea level (and ocean bottom pressure) fluctuation. Sea level budgets can hardly be assessed from such a limited study. Moreover, the paper's conclusion that bottom pressure accounts for 30% of the sea level variance is not demonstrated. In fact, the correlation between ocean bottom pressure and sea level being small in both amplitude and spatial extent, shown in Figure 2, suggests that the former's contribution to the latter, even if it were equivalent to barotropic sea level, is secondary. The correlation's spatial extent does not show the fluctuation being "in unison" across the tropical Indian Ocean either.

In summary, the manuscript is based on false assumptions and questionable claims (e.g., that bottom pressure corresponds to barotropic variation mentioned above). Conclusions are not supported by the evidence provided and are misleading (see also other examples below). The focus of the paper (variations within periods of 30 to 80 days) is also too narrow to be of interest to the oceanographic audience of Nature Communications.

Reply: We have thoroughly revised the manuscript, now explicitly showing that the effect of baroclinicity in the bottom pressure anomaly is negligible (typically inferior to 10%, see previous point) and we believe that it now appears reasonable to assume the intraseasonal bottom pressure anomaly to be a proxy of intraseasonal barotropic SLA during December-April in the tropical Indian Ocean. The fraction of the total standard deviation explained by the intraseasonal barotropic SLA is now explicitly mentioned in the revised manuscript (lines 114-118). It ranges from 30% to 60% in the tropical Indian Ocean, which we believe demands attention. We have also illustrated the time series of intraseasonal barotropic SLA and the total intraseasonal SLA from altimetry in a new figure (Supplementary Fig. 4) to provide a visual experience to the readers about the relative contribution of barotropic SLA at intraseasonal scales. The concern that the focus of the paper on the intra-seasonal timescales is too narrow to be of interest for the readership of the Journal is quite surprising to us, given the vast literature that has been published on the various facets of the MJO (atmosphere, ocean, and their interactions) in Nature portfolio, including Nature Communications and other Journals during the past few years. In particular, the MJO is the most prominent intraseasonal signal in the tropics.

Other Comments:

1) Description of "inverse barometric response" is incorrect. Inverse barometer response is isostatic, meaning changes in atmospheric pressure is compensated by changes in sea level. By definition, there is no bottom pressure variation associated with an inverse barometer response.

Reply: We agree. This is addressed in the revised manuscript. We have rewritten the entire section and removed reference to the inverse barometer effects to avoid confusion, as these are irrelevant for our study.

2) Use of numerous unconventional abbreviations (e.g., BSL, TIO, DJFMA, IBSLA, PSIA, EWD) obfuscate the discussion.

Reply: Usage of abbreviations has been minimized in the revised manuscript.

3) The discussion on spatial scales and their significance are questionable. There are plenty of intra-seasonal variations that exceed 100km in horizontal scale that are not “trapped near the ocean surface” (e.g., equatorial Kelvin waves). Horizontal scales do not inform vertical modes either. For instance, baroclinic variations (e.g., Indian Ocean Dipole Mode) as well as barotropic changes can have scales exceeding 1000km, and hence to say that “coarse-grained SLA results from barotropic processes” is incorrect

Reply: We agree with the Reviewer that a coarse-graining over ~1000 km may not be very efficient in removing the baroclinic signals from observed sea level fields. In the earlier version of the manuscript, we used the coarse-grained altimeter data to show that the response is primarily barotropic. We have thus removed Fig. 2b as well as all reference to coarse-graining in altimetry from the earlier version of the manuscript. This removal however does not alter our inferences. Instead, we have evidenced that the baroclinic contribution to the intraseasonal bottom pressure anomalies is less than 10% in the tropical Indian Ocean and hence the intraseasonal bottom pressure anomalies are primarily modulated by barotropic processes (section 5 in revised methods; Supplementary Fig. 3).

We fully agree that there are large-scale (>100 km) intraseasonal variations that are not trapped near the ocean surface. On the contrary, we argue, in the manuscript, that large-scale wind fluxes can actually break this barrier and carry the response till the ocean bottom thereby inciting a barotropic response. Our point is that these same large-scale wind fluxes will incite baroclinic responses as well.

4) The relationship between wind stress and the Madden-Julian Oscillation (and therefore the significance of MJO to the subject ocean bottom pressure fluctuation) is unclear. For instance, it is not shown how much of the wind stress variability is actually due to the Madden-Julian Oscillation.

Reply: We have cited references that attribute these large anomalous wind stresses over NWAB basin during boreal winter to MJO (line 145).

5) The model correlation (Figure 4) has much larger amplitude with wider spatial extent than do observations (Figure 2). The model does not “reproduce” observations as claimed.

Reply: We believe that the model qualitatively reproduces the observations. One thing is that one may not expect an ocean general circulation model, prone to intrinsic errors, to exactly reproduce the correlation pattern. Beyond this, the correlation pattern obtained from spaceborne gravimetry may also not be considered as absolute truth, in particular as far as the small scales are concerned, given the limited spatial resolution of GRACE state-of-the-art products (see e.g. the reference cited in the paper for a thorough assessment of spaceborne gravimetry).

6) The model experiment with spatially confined forcing (experiment NWAB) has less than half the amplitude of the one with full forcing (UV) (Figure 4), which suggests that most of the relevant forcing exist elsewhere. Therefore, the conclusion that the variations observed at the bottom pressure recorders are driven by “wind stress over the Eastern Indian Ocean” is suspect.

Reply: We agree. This comment is in line with Point#3 of the first Reviewer and Point#2 of the second Reviewer. We have revised thoroughly our results concerning NWAB experiment and other sensitivity experiments. It turns out that the model forcing strategy we had used for the sensitivity experiments (including NWAB experiment) in the previous version was corrupted by an erroneous formulation of the wind-speed - to - wind-stress conversion in the CORE bulk formulae, yielding an underestimation of the wind stress applied to the model by a factor of ~ 2 , compared to our UV reference experiment (which was correct, and has not been changed in the revised version). This mistake happened because we had computed off-line the wind stress (from wind speed) for all the regional sensitivity experiments, based on a too low non-dimensional drag coefficient ($C_d=0.0012$), inconsistent with the CORE bulk formulation embedded in the model and used on-line for the UV experiment. We have now used the same CORE bulk formulation throughout all experiments, including the regionally-forced sensitivity experiments. This bulk formulation increases the wind stress by a factor of ~ 2 in all sensitivity experiments compared to the earlier version. Consequently, the amplitude of barotropic SLA in the NWAB experiment has also considerably increased. The NWAB experiment now reproduces the UV experiment (and hence the observations) to a large extent, particularly during strong MJO events (see the green and red curves in the new Fig. 4b of the revised manuscript). Therefore, we conclude that the NWAB region appears as the primary forcing region for the barotropic fluctuations observed in the tropical Indian Ocean, suggesting that the contribution of the forcing from the other regions is relatively small, particularly during strong MJO events. That is now specifically mentioned in the Methods (subsection “Experiments from MOM4p1”). We believe that the new results are much more convincing than in the earlier version, and that they address the concern of the Reviewer.

7) Ocean circulation is continuous and, as such, a “quantum” of flux makes no sense. The waves referred to in the manuscript are neither demonstrated in the study nor are they evident in the

animation (“video”) provided as supplemental material. A barotropic planetary Rossby wave that travels 45 m/s would have a period of no more than 10-days, not the 30 to 80 days found in the observations. The animation shows the Indian Ocean changing its net mass without compensating changes elsewhere within the plotted domain. It is argued that the mass exchange occurs remotely with the central Pacific Ocean (Figure 6), but the explained waves originate in the Indian Ocean and not the Pacific. There is a logical disconnection in the explanation.

Reply: We agree and have deleted the word "quantum" from the revised manuscript. The barotropic Rossby waves with wavelengths > 2000 km can indeed propagate at about 50 m.s⁻¹ at intraseasonal time scale as can be ascertained from the dispersion relation of planetary barotropic Rossby waves. In the revised manuscript, we have confirmed this objectively in our model simulations, through 2D Radon Transform (provided in Supplementary Fig. 8). Fig. 4c shows that the barotropic SLA over the whole tropical Pacific Ocean is anti-correlated with that at BP-BoB in the Indian Ocean. Hence the hypothesis that the mass change in Indian Ocean is compensated by the Pacific Ocean, looks probable. This is further supported by Fig. 6. Even though the waves originate in the Indian Ocean, they can induce an inter-basin mass flux with Pacific. There are two possible pathways for such a mass exchange: through Indonesian Straits and/or through the Southern Ocean (to the south of Australian sub-continent). The exact pathway of the inter-basin mass fluxes however needs to be investigated.

8) Removing intra-seasonal barotropic variation is not a “standard” approach in satellite data processing. One often does use “2D barotropic models” when utilizing satellite data, but only to remove high frequency variations that can otherwise be aliased. The Nyquist frequency for the constellation of satellite sea level measurements in operation today is less than 10-days, which is far shorter than the 30 to 80-day period analyzed in this study. Effects of atmospheric pressure loading that deviate from inverse barometer response is generally considered to be shorter than a few days, again much shorter than the period analyzed here. None of these approaches “miss a large fraction of the actual signal.”

Reply: We agree. We have removed this mention from the manuscript. In fact, we have completely re-written the whole "Discussions and Implications" section, to make it more focussed and more relevant.

9) Barotropic Rossby waves was said to have a speed of 45 m/s, which takes 2-days to cross the Indian Ocean. This cannot possibly adjust the Indian Ocean in a “few hours” as stated in “Discussion and Implications”. The MJO response discussed earlier was also in terms of barotropic waves, not baroclinic variability.

Reply: We agree. We have changed "few hours" to "few tens of hours" in the revised manuscript (lines 318-320).

Reviewers' comments:

Reviewer #1 (Remarks to the Author):

The authors have made major changes to the manuscript in response to the extensive comments from the reviewers. In regard to the main points raised in my original review, the responses are mostly satisfactory. I clarify below what changes I think are still needed to render the paper publishable. I see these as relatively minor revisions, which the authors should be able to carry out without much problem.

MAIN POINTS

#1R. Regarding my original point #3, I think the authors still overemphasize the importance of the winds in the box in Fig 3a. This is evident from the considerable differences in the NWAB experiment (green curve, Fig 4b) compared to the full forcing experiment (red curve, Fig 4b). A more quantitative test can be done by calculating the variance in the red curve R explained by the green curve G as

Variance in R explained by G = $1 - (\text{variance}[G - R]) / \text{variance}[G]$ (multiply by 100 to get result in %)

Such diagnostic can set the tone for the discussion and what can be claimed by the authors.

#2. The use of "accounts for" in line 19 is misleading. A more accurate statement would be something like "significant basin-wide barotropic sea level variability in the tropical Indian Ocean during December-April, with standard deviations amounting to ~30-60% of the standard deviation in total intraseasonal sea level." Similarly, in line 115 and line 305, you should use "amounts to" instead of "explains". To address explained variability, one would need analysis that consider not only amplitude (as done here) but also phase.

#3. Key to the assessment of the barotropic nature of the variability discussed is the large scale coherence of the BPRs. For this reason, I think the inclusion of supplementary fig 2 in the main text (perhaps as extra panels in fig 1) would be warranted. More importantly, the discussion in the text needs to reflect some of the ambiguity in those coherence results (e.g., higher, significant coherence does not necessarily align always with the December-April period, the BP-BoB and BP-NWAB records are largely incoherent after 2013). Most of this is also apparent in the filtered time series seen in fig 2a. The message in the text must be one of, yes, results are suggestive of coherent behavior but contain considerable noise.

OTHER COMMENTS AND SUGGESTED EDITS (BY LINE #)

15. Rewrite as "...barotropic processes (involving the entire water column) or baroclinic processes (governed by stratification)."

17. Add "(periods of 30-80 days)" after "time scales".

23. Rewrite as "...prominent role in intraseasonal sea level and mass budgets." It is a bit of stretch to say they play a prominent role in momentum budget, as barotropic currents are likely weak, compared to surface-intensified baroclinic currents.

36. Delete "other than astronomical tides".

58. Should be "Observations of bottom pressure variability"?

59. Rewrite as "...pressure, which is expressed...".

103-105. There are various misalignments in phase. Also last statement is not strictly true: one could have baroclinic signals agree at the large scale if forcing was coherent on the large scale.

105-106 Delete "Since gravimetry is sensitive only to barotropic SLA variability". Gravimetry will measure bottom pressure, regardless of its barotropic or baroclinic origins.

141-143. Statement is most obscure to me.

321. "...realistically simulate..."

340-348. Largely speculative and vague. Should be deleted.

Methods 110-112. Does not make sense. GRACE and BPR measure essentially the same field.

Reviewer #2 (Remarks to the Author):

I have read through the revised manuscript and found it improved from the previous version. However, I still have questionable points about the sensitivity experiments using an OGCM.

(1) The purpose for the sensitivity experiments of NWAB-FB is not clear. Since the flat bottom is applied for all the region in this experiment, topography effects are all neglected. This also modify the source term in the northwestern Australian basin, if I understand correctly, causing significant reduction of the ocean responses to the same wind forcing applied to the other experiments with the bottom topography. Therefore, the results of NWAB-FB include changes in both the forcing and subsequent propagating characteristics. This obscure interpretation of the results and not as simple as the authors addressed in the text. Even if the topography within the forcing region remains but the flat bottom elsewhere, the perimeter of forcing region provide artificial topographic change, thereby causing artificial baroclinic-barotropic conversion. This also distorts result in the model response. We cannot simply say that the differences between NWAB and NWAB-FB are due to baroclinic-barotropic conversion.

(2) Using results from NWAB-NR experiment, the authors claim the importance of the Southeastern Indian Ridge to reduce the signal in the Bay of Bengal (p10, line 239). How is this conclusion drawn? There are many other topographic features that affect propagation of barotropic waves and these features are also excluded in this experiment. The arguments here are rather subjective to have several conclusive sentences, one of them is shown above. Sensitivity experiments should be designed to clarify only one aspect of interests, otherwise results tell us nothing.

(3) In the section "Dynamics of Barotropic SLA", three kinds of wave propagations are proposed as the important processes affecting the basin-scale barotropic response to the intraseasonal wind forcing. Degree of importance of the three waves seems not to be the same, but this point is not argued in the paper. I believe that the NWAB-CF experiment is designed to address a part of this issue. Discussions should be included.

Minor point

P10, line 230-231: the pink arrow in Fig.5 does not originate from the southwest corner of the box.

Point by point replies to the Reviewers comments (Manuscript number: NCOMMS-18-08672A-Z)

Reviewer #1

Main Points

#1. Regarding my original point #3, I think the authors still overemphasize the importance of the winds in the box in Fig 3a. This is evident from the considerable differences in the NWAB experiment (green curve, Fig 4b) compared to the full forcing experiment (red curve, Fig 4b). A more quantitative test can be done by calculating the variance in the red curve R explained by the green curve G as Variance in R explained by G = $1 - (\text{variance}[G - R]) / \text{variance}[G]$ (multiply by 100 to get result in %).

Such diagnostic can set the tone for the discussion and what can be claimed by the authors.

Reply:

We agree that the explained variance is more informative than the simpler statistics we had provided in the earlier version. The variance of barotropic SLA in the UV experiment (our reference experiment, red curve of Fig. 4b) explained by the NWAB experiment (green curve of Fig. 4b) at BP-BoB amounts to ~60%, which is not a small value. The formula used is:

Variance in red curve (R) explained by green curve (G) = $[1 - (\text{variance}(G-R))/\text{variance}(R)] \times 100$.

We included this diagnostic in the manuscript as suggested by the Reviewer (line 197-199).

#2. The use of “accounts for” in line 19 is misleading. A more accurate statement would be something like “significant basin-wide barotropic sea level variability in the tropical Indian Ocean during December-April, with standard deviations amounting to ~30-60% of the standard deviation in total intraseasonal sea level.” Similarly, in line 115 and line 305, you should use “amounts to” instead of “explains”. To address explained variability, one would need analysis that consider not only amplitude (as done here) but also phase.

Reply:

We agree. We have changed the wording accordingly. (line number 22-24; line 129-130 and line 338).

#3. Key to the assessment of the barotropic nature of the variability discussed is the large scale coherence of the BPRs. For this reason, I think the inclusion of supplementary fig 2 in the main text (perhaps as extra panels in fig 1) would be warranted. More importantly, the discussion in the text needs to reflect some of the ambiguity in those coherence results (e.g., higher, significant coherence does not necessarily align always with the December-April period, the BP-BoB and BP-NWAB records are largely incoherent after 2013). Most of this is also apparent in the filtered time series seen in fig 2a. The message in the text must be one of, yes, results are suggestive of coherent behavior but contain considerable noise.

Reply:

Now we have included the supplementary Fig. 2 as an extra panel in Fig.1 (bottom panel). We have also included the statements on this in the revised manuscript (line number 84-94). The statements highlight the reasons for the poor coherence of the signals at different locations, after 2013.

Minor points

15. Rewrite as "...barotropic processes (involving the entire water column) or baroclinic processes (governed by stratification)."

Reply:

Modified as suggested (line 18).

17. Add "(periods of 30-80 days)" after "time scales".

Reply:

Modified as suggested (line 20).

23. Rewrite as "...prominent role in intraseasonal sea level and mass budgets." It is a bit of stretch to say they play a prominent role in momentum budget, as barotropic currents are likely weak, compared to surface-intensified baroclinic currents.

Reply:

We agree. Modified as suggested (line 27).

36. Delete "other than astronomical tides".

Reply:

Deleted as suggested (line 38).

58. Should be "Observations of bottom pressure variability"?

Reply:

We acknowledge that the section deals with the observations of bottom pressure variability. But our main objective of this study is to show that there are significant basin-wide barotropic fluctuations in the intraseasonal sea level in Tropical Indian Ocean. As we have shown, bottom pressure anomaly, in this work, is a good proxy of barotropic sea level anomaly (see Supplementary Fig. 2 of the revised version). Hence, the contribution of

baroclinic processes is negligible at intraseasonal time scales in the present case. We believe that changing the title of the section to “*Observations of bottom pressure variability*” would unnecessarily divert the attention of the reader and may lead to undue emphasis on bottom pressure. We rather wish to arrest the focus of the reader to barotropic fluctuations and would prefer to stick to the present title, i.e., “*Observations of barotropic sea level variability*”, for clarity.

59. Rewrite as “...pressure, which is expressed...”.

Reply:

Modified as suggested (line 61).

103-105. *There are various misalignments in phase. Also last statement is not strictly true: one could have baroclinic signals agree at the large scale if forcing was coherent on the large scale.*

Reply:

We agree. In principle, slow baroclinic processes cannot be emphatically ruled out if there exists a large-scale intra-seasonal forcing spanning over the large area of tropical Indian Ocean. However, no such large-scale forcing has ever been reported in the Indian Ocean that encompasses all the three BPR locations. For clarity, we have included a short discussion in the revised manuscript explicitly to address this concern (line 116-119).

105-106 *Delete “Since gravimetry is sensitive only to barotropic SLA variability”. Gravimetry will measure bottom pressure, regardless of its barotropic or baroclinic origins.*

Reply:

We agree. We replaced this statement with “To further explore the possibility of basin-wide EWD variability,” (line 119-122).

141-143. *Statement is most obscure to me.*

Reply:

This statement was introduced in the manuscript during the previous round of revision, at the behest of the second Reviewer. His concern was to understand why the wind stress curl appears to have a monopole structure once the wind stress is scaled by $1/H$ (see the formulation of the wind source term in line 143), whereas MJO zonal wind stress curl typically shows a dipolar structure in latitude. We now realise that the single sentence we had added to address the issue of absence of dipolar structure of the wind stress curl during MJO events was insufficient. We have now introduced a couple of additional sentences to convey the message more clearly (line 155-159). We have also directed the

readers to the two videos (video 1 and video 3) wherein it can be seen that the dipolar structure in the wind stress curl collapses once gradients in topography is introduced.

321. *"...realistically simulate..."*

Reply:

Modified as suggested (line 354).

340-348. *Largely speculative and vague. Should be deleted.*

Reply:

We agree that the last line of the paragraph, viz., "This may aid in catalyzing instabilities in the vicinity of boundary currents." is speculative in nature. Hence the statement has been deleted in the revised version. However, to point out one of the consequences of this study, we have added a modified statement "We can expect that" at line 373. Also, we have retained the rest of the paragraph, as we believe that this work will lead to relevant and necessary research on those aspects. Indeed, ever since the seminal study of Wunsch (JPO 1997), it is known that the barotropic mode accounts for a large, if not dominant, fraction of the depth-integrated kinetic energy of the tropical and subtropical oceans. We have explicitly computed the barotropic kinetic energy contained in the north Indian Ocean basin at intraseasonal time scale in the NWAB experiment. The peak value is ~ 20 peta Joules. To put it in perspective, the global mean barotropic kinetic energy is ~ 500 peta Joules (Aiki et al., 2011; <https://doi.org/10.1007/s10236-011-0382-y>). We believe that 20 peta Joules at intraseasonal time scales is significant in terms of magnitude and it is of consequence because it has its origin in a relatively small area (compared to the global ocean area) located in north-west Australian Basin. Hence, we believe that it is prudent to point out that the energy might act as a reservoir for deep ocean intraseasonal variabilities. In line with the high magnitude of barotropic kinetic energy, we had also computed the mean vertically integrated velocity of the flow during December-April and it turns out to be non-negligible $\sim 2 \text{ cms}^{-1}$. This is stated in paragraph (line 374-376). We believe this is also significant and is not irrelevant in the context of the section on "Discussions and Implications".

Methods 110-112. Does not make sense. GRACE and BPR measure essentially the same field.

Reply:

We agree. We have modified the statement in the revised Supplementary Methods (line number 110). There are now no redundant reference to either GRACE or BPR under that section.

Reviewer #2

(1) The purpose for the sensitivity experiments of NWAB-FB is not clear. Since the flat bottom is applied for all the region in this experiment, topography effects are all neglected. This also modify the source term in the northwestern Australian basin, if I understand correctly, causing significant reduction of the ocean responses to the same wind forcing applied to the other experiments with the bottom topography. Therefore, the results of NWAB-FB include changes in both the forcing and subsequent propagating characteristics. This obscure interpretation of the results and not as simple as the authors addressed in the text. Even if the topography within the forcing region remains but the flat bottom elsewhere, the perimeter of forcing region provide artificial topographic change, thereby causing artificial baroclinic barotropic conversion. This also distorts result in the model response. We cannot simply say that the differences between NWAB and NWAB-FB are due to baroclinic-barotropic conversion.

Reply:

We agree. As stated in the manuscript (line number 209-210), the purpose of the NWAB-FB experiment was to identify the overall role of topography. The effect of topography can indeed manifest through various ways, as rightly pointed out by the Reviewer. First, the NWAB-FB experiment allowed us to identify the presence of topographic waves (ruling out coastally-trapped Kelvin waves, see line number 303-304). Second, we used NWAB-FB experiment to assess the role of the baroclinic-barotropic conversions. We agree with the Reviewer that by implementing a flat bottom of depth 3000 m everywhere, the magnitude of the wind forcing term will be changed, and consequently our inference about baroclinic-barotropic conversion may not be fully justified. We have therefore performed an additional flat bottom sensitivity experiment (NWAB-FB-SWS) wherein we have scaled the wind stress by $H_0/H(x,y)$ so that the rate of vorticity input to the ocean remains unchanged, identical to the NWAB experiment with actual topography. Here, $H(x,y)$ is the actual ocean depth and H_0 is the uniform depth of 3000 m set in the flat bottom experiment (NWAB-FB-SWS). In this NWAB-FB-SWS experiment, any changes in barotropic SLA compared to NWAB experiment can therefore be attributed solely to the absence of baroclinic-barotropic conversion, and not to changes in the wind source term. It was found that the model response remains similar to what it was in NWAB-FB experiment (Figure R1, top panel). This proves that the reduction of the amplitude of intraseasonal barotropic SLA variability in the experiment with flat bottom (NWAB-FB) is prominently explained by the lack of baroclinic-to-barotropic conversion facilitated by gradients in topography. We have therefore added a couple of sentences in the revised manuscript and a figure in the Supplementary Figures (Supplementary Fig. 6; also in the top panel of Figure R1) to inform the reader that even if the rate of vorticity input is unchanged in a flat bottom experiment, the barotropic response at BP-BoB remains essentially the same (line number 218-221).

We see from the time series of intraseasonal barotropic SLA at BP-BoB from NWAB (black curve) and NWAB-NR (red curve) experiment that the absence of topographic gradients outside the forcing region causes a reduction in the amplitude by $\sim 23\%$ (Figure R1, bottom panel). However, the reduction in amplitude of intraseasonal barotropic SLA from

NWAB-FB-SWS (blue curve) experiment with respect to NWAB experiment ranges from 33%-45%. The difference of 10-22% can be attributed to the absence of topographic gradients inside the forcing region because the input vorticity to the ocean remains same for NWAB, NWAB-NR and NWAB-FB-SWS experiments.

Regarding the Reviewer's concern about the potential role of topographic gradients outside the forcing region, it can be ascertained from the NWAB-NR experiment where the topography outside the forcing zone was gradually flattened over a length scale of 500 km. The gradual flattening over such a broad distance implies very weak bottom slope, and mitigates possibilities of significant baroclinic-barotropic conversions in the perimeter zone. This is now explicitly mentioned in the revised manuscript (line 258-262).

Figure R1 | (top panel) Time series of intraseasonal barotropic SLA obtained from NWAB-FB experiment (black) and NWAB-FB-SWS experiment (red) at BP-BoB during the period December 2010 – April 2015. $y=0$ line is plotted for clarity. (bottom panel) Time series of intraseasonal barotropic SLA obtained from NWAB (black), NWAB-NR (red) and NWAB-FB-SWS (blue) at BP-BoB during the period December 2010 – April 2015. $y=0$ line is plotted for clarity.

(2) Using results from NWAB-NR experiment, the authors claim the importance of the Southeastern Indian Ridge to reduce the signal in the Bay of Bengal (p10, line 239). How is this conclusion drawn? There are many other topographic features that affect propagation of barotropic waves and these features are also excluded in this experiment. The arguments here are rather subjective to have several conclusive sentences, one of them is shown above. Sensitivity experiments should be designed to clarify only one aspect of interests, otherwise results tell us nothing.

Reply:

We agree with the observations made by the Reviewer. In order to address this concern, we have conducted two additional sensitivity experiments. In the first sensitivity experiment, we have removed the Ninety East Ridge (NWAB-noNER experiment). The rest of the topographic features in the entire Indian Ocean domain remain unaltered. The forcing is confined only within the NWAB region. Figure R2a, shows the intraseasonal barotropic SLA from the NWAB experiment and the NWAB-noNER experiment at BP-BoB. The NWAB-noNER experiment reproduces the variability of the NWAB experiment, but the signal reaches BP-BoB almost ~ 1 day (Figure R2b) later than in the NWAB experiment. This implies that the Ninety East Ridge plays a role in steering the planetary Rossby wave north-westward (line number 368), though the overall impact on the intraseasonal variability of barotropic SLA in terms of amplitude in the northern Indian Ocean is weak. We modified the manuscript to explain this clearly (line number 275-277). Ninety East Ridge is located just outside the western edge of the forcing region. Flattening the ridge smoothens strong topographic gradients on the perimeter of the western edge of the box. And yet there is negligible change in amplitude. This suggests that the topographic features outside the western perimeter play a negligible role in baroclinic-barotropic conversions.

Figure R2 | (a) Time series of intraseasonal barotropic SLA obtained from NWAB (black) and NWAB-noNER (red) at BP-BoB during the period December 2010 – April 2015. (b) close up view of (a) during the period Feb 01-April 30, 2012. $y=0$ line is plotted for clarity.

In the second sensitivity experiment, conversely, we removed the Southeast Indian Ridge instead of the Ninety East Ridge (NWAB-noSEIR experiment). Figure R3 shows the time series of intraseasonal barotropic SLA from NWAB experiment (black curve) and NWAB-

noSEIR experiment (red curve) in the subtropics at 120° E, 50° S – a location to the south of Southeast Indian Ridge (Figure R3). It is apparent that the amplitude of the variability almost doubles due to the absence of the ridge, i.e., the topographic wave leaks to the south in the absence of Southeast Indian Ridge. Alternatively, the Southeast Indian Ridge acts as a barrier and contains the barotropic fluctuations to its northern flank. We have also estimated the effect of the absence of Southeast Indian ridge at the BP-BoB location. The amplitude reduces by 12-16% if this ridge is absent. We recall that if all topographic features outside the forcing region are absent (NWAB-NR experiment), the amplitude reduces by $\sim 23\%$ (line number 264-268). We have therefore made the appropriate modifications in the revised manuscript (line number 268-271).

Figure R3 | Time series of intraseasonal barotropic SLA obtained from NWAB (black) and NWAB-noSEIR (red) at 120° E, 50° S during the period December 2010 – April 2015. $y=0$ line is plotted for clarity.

(3) In the section “Dynamics of Barotropic SLA”, three kinds of wave propagations are proposed as the important processes affecting the basin-scale barotropic response to the intraseasonal wind forcing. Degree of importance of the three waves seems not to be the same, but this point is not argued in the paper. I believe that the NWAB-CF experiment is designed to address a part of this issue. Discussions should be included.

Reply:

We agree that more clarity is needed on this issue in the manuscript. The planetary wave is primarily responsible for producing the observed fluctuations in the bottom pressure records – particularly at the BP-BoB and the BP-AS locations. When the planetary wave is absent (NWAB-CF experiment), $\sim 95\%$ of the amplitude is not reproduced at the BP-BoB location compared to NWAB experiment. This is consistently true throughout the northern and central Indian Ocean. The southern tropical Indian Ocean is however influenced by the topographic Rossby wave. The combined effect of these two waves is to set up a basin-wide fluctuation of the tropical Indian Ocean that extends from the northern boundary of the Indian Ocean to the Southeast Indian Ridge in the south. The continental shelf wave

appears to take no direct part in the basin-wide fluctuations. We have now introduced a discussion on this in the revised manuscript (line number 292-296; 316-320).

Minor point

P10, line 230-231: the pink arrow in Fig.5 does not originate from the southwest corner of the box

Reply:

We agree. We have re-examined the video (video 1) more carefully. The topographic Rossby wave does not actually originate from the southwest corner of the box as marked in the previous version of the manuscript. Rather it appears to originate from the central part of the box. We have therefore deleted the statement on the wave originating from the south-west corner from the revised manuscript. Also, we have made sure that the pink arrow originates from the center of the box in the schematic diagram (Fig. 5) in the revised manuscript.

REVIEWERS' COMMENTS:

Reviewer #1 (Remarks to the Author):

The authors have satisfactorily answered the points mentioned in my last review and I recommend publication at this time.

Reviewer #2 (Remarks to the Author):

The revised manuscript addressed all of my comments appropriately, and now all the contents are understandable and show interesting results, which merit publication.

I think the manuscript can be accepted now.

Very minor one point: in Supplementary Figure 6, the red curve is indicated as "NWAB-FB-SWS", but I am not sure which case is this and what "SWS" stands for.